

# Post-depositional modification on seasonal-to-interannual timescales alters the deuterium excess signals in summer snow layers in Greenland

Michael S. Town[1,2], Hans Christian Steen-Larsen[2], Sonja Wahl[2,3], Anne Katrine Faber[2], Melanie Behrens[4], Tyler R. Jones[5], and Arny Sveinbjornsdottir[6]

[1]Earth and Space Research, Seattle, USA
[2]Geophysical Institute and Bjerknes Centre for Climate Research, University of Bergen, Bergen, Norway
[3]École Polytechnique Fédérale de Lausanne, Lausanne, Suisse
[4]Alfred-Wegener-Institut Helmholtz-Zentrum für Polar- und Meeresforschung, Research Unit Bremerhaven, Bremerhaven, Germany
[5]Institute of Arctic and Alpine Research, University of Colorado, Boulder, USA
[6]Institute of Earth Sciences, Reykjavík, Iceland

**Correspondence:** Michael S. Town (michael.town@esr.org)

**Abstract.** We document the isotopic evolution of near-surface snow at the EastGRIP ice core site in the Northeast Greenland National Park using a time-resolved array of 1-m deep isotope ($\delta^{18}O$, $\delta D$) profiles. The snow profiles were taken from May-August during the 2017-2019 summer seasons. An age-depth model was developed and applied to each profile mitigating the impacts of stratigraphic noise on isotope signals. Significant changes in deuterium excess ($d$) are observed in surface snow and

near-surface snow as the snow ages. Decreases in $d$ of up to 5 $‰$ occurs during summer seasons after deposition during two of the three summer seasons observed. The $d$ always experiences a 3-5 $‰$ increase in $d$ after aging one year in the snow due to a broadening of the autumn $d$ maximum. Models of idealized scenarios coupled with prior work (Wahl et al., 2022) indicate that the summertime post-depostional changes in $d$ ($\Delta d$) can be explained with surface sublimation, forced ventilation of the near-surface snow down to 20-30 cm, and isotope-gradient-driven (IGD) diffusion throughout the column. The interannual

$\Delta d$ is also partly explained with IGD diffusion, but other mechanisms are at work that leave a bias in the $d$ record. Thus, $d$ does not just carry information about source region conditions and transport history as is commonly assumed, but also integrates local conditions into summer snow layers as the snow ages. Finally, we observe a dramatic increase in the seasonal isotope-to-temperature sensitivity occurs, which can be explained solely by IGD diffusion. Our results are dependent on the site characteristics (e.g. wind, temperature, accumulation rate), but indicate that more process-based research is necessary to

understand water isotopes as climate proxies. Recommendations for monitoring and physical modeling are given, with special attention to the $d$ parameter.

## 1 Introduction

The relative concentration of stable water isotopes in polar snow and ice have proven useful in temperature reconstructions of Earth's climate (e.g., Lorius et al., 1990; Jouzel et al., 1997; Johnsen et al., 2001; Jouzel et al., 2003; Kavanaugh and Cuffey,



**Table 1.** Terms used in this study to describe different sources of solid water isotope.

| Ice type | description |
| --- | --- |
| Precipitation | ice particles caught and measured before they hit the ground |
| Surface snow | ice particles collected from the surface down to 1 cm |
| Near-surface snow | ice particles from 1 cm - 100 cm |

2003; Steig et al., 2013). In the past, these reconstructions were dependent on understanding the sensitivity of changes in water isotopes in polar snow and ice to changes in mean annual temperature in the polar regions, i.e., the water-isotope-temperature sensitivity. Small changes in this sensitivity have significant influence on inferences about past climates based on polar ice cores (e.g., Grootes et al., 1993; Charles et al., 1994; Petit et al., 1999; Jouzel et al., 2003). Recent climate reconstruction efforts are not as dependent on temperatures inferred from water isotopes in polar snow because they use proxies other than

water isotopes in polar snow to understand regions outside of the poles (e.g., Rohling et al., 2012; Dahl-Jensen et al., 2013; Buizert et al., 2021). Simulations of past polar ice sheet mass balance and climate still require accurate knowledge of ice sheet temperatures often derived from empirical isotope-temperature sensitivities (e.g., Cuffey et al., 2016; Jones et al., 2023), or more nuanced meteorological approaches involving regional temperature gradients and patterns in transport (e.g., Markle and Steig, 2022). Inferences about past circulation and weather patterns are also possible from combinations of isotope and other

chemistry measurements from polar snow and ice (e.g., Mayewski et al., 1994; Steffensen et al., 2008; Guillevic et al., 2013; Jones et al., 2018). Such understanding is important not only to make claims about past climate, but to improve models for prediction of weather and future climate (e.g., Blossey et al., 2010; Werner et al., 2011; Dee et al., 2015; Dütsch et al., 2019).

Despite the importance of isotope signals in polar snow and ice to understanding climate and temperature, there remains a lack of contiguous understanding from source to extraction of the integrated relationship between local and regional climate and

the isotopic composition of polar snow. Specifically, there is uncertainty about what happens to the isotopic signal in the top meter of snow when it is still under the influence of local meteorology. This study provides observations of meteorology-induced isotopic changes in surface and near-surface snow. Subsequent analysis and modeling provoke some revised interpretation of the $d$ climate proxy.

### 1.1 Water isotopes in the atmospheric hydrologic cycle

The part of the global hydrologic cycle that brings precipitation to the polar regions provides several opportunities for isotopic fractionation. The relative isotopic content of the precipitation (eq. 1) is therefore thought to represent an integrated history of the water from source to sink (Craig, 1961; Dansgaard, 1964; Gonfiantini, 1978). It is important to clarify here that prior literature relating to water isotopes in polar often uses the terms precipitation, surface snow, and near-surface snow interchangeably. It is appropriate for our thesis to keep these terms distinct; our usage is outlined in Table 1. We apply these same terms to past

literature no matter what term was used in the original literature.





So, in referring to historical work we use the term we think most appropriate for that work regardless of what term they used. The critical distinctions here are if the it *precipitation*. If particles were collected from the surface, no matter how long it likely rested there, we refer to is as *surface snow*. Snow that has been extracted from below 1 cm is called *near-surface snow*.

$$\delta^{18}O = (\frac{\frac{H_2^{18}O}{H_2^{16}O}sample}{\frac{H_2^{18}O}{H_2^{16}O}standard} - 1) \tag{1}$$

Water isotopes in the global hydrologic cycle have been monitored extensively since the 1960s, illustrating robust linear relationships between $\delta^{18}O$ and $\delta D$ (eq. 2). The y-intercept of this relationship is commonly referred to as 'deuterium excess' (d-excess or $d$, equation 3; e.g., Dansgaard, 1964; Merlivat and Jouzel, 1979; Jouzel and Merlivat, 1984). A mean $d$ value of approximately 10 $^o/_{oo}$ for global precipitation is thought to represent equilibrium fractionation conditions (Dansgaard, 1964). The $d$ parameter is often used as an integrated characterization of an air mass's hydrologic source and transport history (Mer-
livat and Jouzel, 1979; Jouzel and Merlivat, 1984; Ciais and Jouzel, 1994; Pfahl and Sodemann, 2014; Hu et al., 2022). The mean northern hemisphere $d$ seasonal cycle has a maximum in winter, and minimum in summer from hemispherically averaged Global Network of Isotopes in Precipitation stations (Pfahl and Sodemann, 2014). However, Johnsen and White (1989) observed an autumn peak and spring minimum in $d$ from near-surface snow. Kopec et al. (2022) recently observed a summertime peak in $d$ in precipitation at Summit, Greenland, possibly shifted from autumn due to influence of upwind ice sublimation. Sim-
ilar discrepancies exist in Antarctic records. Delmotte et al. (2000) show a $d$ seasonal cycle in shallow cores from the coastal Law Dome site in East Antarctica that peak in winter and have a minimum in the autumn and Summer. However, Schlosser et al. (2008) show a more complicated $d$ signal exists when considering snow with minimal exposure to post-depositional effects. Through back trajectory compositing, they show that moisture with an oceanic origin has a maximum $d$ in winter and a minimum in summer.

$$\delta D = 8 \cdot \delta^{18}O + 10 \tag{2}$$

$$d = \delta D - 8 \cdot \delta^{18}O \tag{3}$$

Linear relationships between mean annual surface temperature and water from precipitation or near-surface snow (a.k.a. isotope-temperature sensitivity) have also been defined using spatially-distributed measurements ($\gamma_s$, see equation 4, Dansgaard, 1964). Often different linear relationships exist for similar areas when looking at temporally-oriented data sets and
models ($\gamma_t$, e.g., Cuffey et al., 1995, 2016; Werner et al., 2018).

$$\gamma = \frac{\Delta\delta^{18}O}{\Delta T} \tag{4}$$





Improved modeling of the hydrologic cycle and cloud physics are a primary focus of current isotope-enabled models (IEMs) with a range of complexity, which has improved interpretation of snow and ice cores (e.g., Merlivat and Jouzel, 1979; Jouzel and Merlivat, 1984; Johnsen and White, 1989; Ciais and Jouzel, 1994; Blossey et al., 2010; Werner et al., 2011; Markle and Steig, 2022). Some focus is still on water-isotope-temperature relationships like $\gamma_t$ (e.g., Werner et al., 2018). Yet, it is recognized that more comprehensive, process-based approach to isotope-climate relationships using trajectory mixing, source-to-sink temperature gradients, and non-linear isotope-to-temperature sensitivities is necessary due to the complexity of integrated processes leading up to deposition (e.g., Markle and Steig, 2022).

A challenge for all climate-to-isotope relationships and IEMs is validation. These relationships and IEMs are compared, or even tuned, to surface and near-surface snow that was treated as if it were precipitation (e.g., Jouzel and Merlivat, 1984; Johnsen and White, 1989; Petit et al., 1991; Uemura et al., 2012; Werner et al., 2018; Dütsch et al., 2019; Markle and Steig, 2022), but the snow has in fact spent months or years exposed to impacts of post-depositional processes. The water isotope signal will have most certainly changed after deposition due to local meteorology-induced snow metamorphism. These changes then are inadvertently and inappropriately integrated into 'before deposition' mechanics of isotope relationships and IEMs.

## 1.2 Isotopic evolution after deposition

After deposition at a polar site, the isotopic content of snow continues to evolve in response to its surrounding environment. Diffusion along isotopic gradients is considered a dominant process from 2 m below the snow surface to firn close-off, along with advection and thinning (Johnsen, 1977; Johnsen et al., 2000; Gkinis et al., 2014; Jones et al., 2017), with other atmospheric-driven processes being irrelevant below this depth. Proper inversion of these processes are necessary for accurate reconstruction of timing and magnitude of isotopic signals at frequencies affected by diffusion, usually in the range of seasonal-to-decadal scales (e.g., Johnsen et al., 2000; Vinther et al., 2010; Jones et al., 2018, 2023).

While necessary, inversion of IGD diffusion is not always sufficient to reconstruct $\delta^{18}O$ or $d$ at the time of deposition. For example, observations at Dome Fuji, Antarctica show a disconnect between the magnitude of the $\delta^{18}O$ annual cycle in precipitation and the firn that cannot be reconciled through inversion of IGD diffusion (Fujita and Abe, 2006). Other post-depositional processes like wind-driven mixing (e.g., Fisher et al., 1985; Kochanski et al., 2018), atmosphere-surface exchange (Steen-Larsen et al., 2014; Hughes et al., 2021; Wahl et al., 2021, 2022), or snow metamorphism (e.g., Ebner et al., 2017) are also likely influencing these isotopic signals. Modeling studies have shown that local meteorology can smooth and bias isotope records by imprinting near-surface atmospheric water vapor isotopic signals in the near-surface snow through forced-ventilation (i.e. wind pumping, Waddington et al., 2002; Neumann and Waddington, 2004; Town et al., 2008b). The resulting isotopic bias is predicted to occur during the relatively warmer summers in isotopically depleted winter layers at low accumulation sites (Town et al., 2008b).

What to do about these influences to refine more accurate climate signals from water isotopes is an area of current concern. Simply averaging horizontally across wind-induced snow structures causes large variability in environmental signals which may unnecessarily combine distinct layers (e.g., Steffensen, 1985; Münch et al., 2017; Zuhr et al., 2021, 2023). Increased fidelity in surface and near-surface snow isotope observations have led to improved understanding of the myriad mechanisms





influencing surface and near-surface processes, bringing us closer to mechanistic understanding of post-depositional isotopic modification.

For example, Casado et al. (2021) show observational evidence of post-depositional change in surface snow induced by sublimation/deposition mechanisms, citing insolation and other surface energy budget processes as important to the surface
$\delta^{18}O$ and $d$ signals. Observed changes in surface snow $\delta^{18}O$ at EastGRIP has been successfully simulated by incorporating sublimation into an isotope-enabled surface energy budget model (Wahl et al., 2022). The stable boundary layer (SBL) over high altitude Antarctica likely influences surface isotopic content resulting in enrichment of surface $\delta^{18}O$ at the expense of $\delta^{18}O$ vapor in the SBL (e.g., Ritter et al., 2016; Casado et al., 2018) . Windier sites will have a more well-mixed atmospheric boundary layer, resulting in correlation between surface snow $\delta^{18}O$ content and atmospheric surface layer $\delta^{18}O$ vapor content
(e.g., Steen-Larsen et al., 2014; Wahl et al., 2022). At low accumulation sites scouring and redistribution of annual layers is always a problem to contend with (e.g., Epstein et al., 1965; Casado et al., 2018).

On the other hand, snow pit data from East Antarctica indicate that IGD diffusion, precipitation intermittency, and possibly spatial inhomogeneity may explain isotopic signal to noise ratios, and additional mechanisms are not necessary (Münch et al., 2017; Laepple et al., 2018). At Summit Station, Greenland, Kopec et al. (2022) found very little post-depositional change in
isotopic content of precipitation or near-surface snow after deposition, and also indicate that upwind sublimation is responsible for the unique isotopic signatures observed in precipitation at Summit Station. Town et al. (2008b) show that the high accumulation rate (24 cm a$^{-1}$ l.w.e.) and relatively warm mean annual temperature at Summit Station are competing influences in isotopic post-depositional modification. Looking at one summer season at EastGRIP (summer 2019), Zuhr et al. (2023) find evidence of local processes inducing post-depositional change in $d$, but no change in $\delta^{18}O$, in snow down to 10 cm, with the
interannual consistency and potential causes remaining unexplored.

## 1.3 This study

To investigate discrepancies in evidence and primary mechanisms of post-depositional modification of water isotope content of near-surface snow, we present analysis of a time-resolved surface snow and near-surface snow profile data set from the East Greenland Ice Core Project (EastGRIP) site in Northeast Greenland (Mojtabavi et al., 2020). Our study asks the following
questions:

– What is happening to the water isotope signal at the snow surface and in the near-surface snow at EastGRIP, while the snow is still within the dynamic influence of the local atmosphere?

– Can any changes in isotopic content ($\delta^{18}O$, $\delta D$, $d$) observed at EastGRIP be explained by existing theory or models?

To answer these questions, we collected and analyzed arrays of overlapping 1-m snow cores during three summer field
seasons (2017-2019) at the EastGRIP ice core site. The snow spans the time period 2014-2019. Analyzed for water isotopic content and indexed to an age-depth model, the resulting data set chronicles isotopic evolution of surface and near-surface snow throughout each summer season, and interannually. The isotope data set is supported by meteorology from the PROMICE



network (Fausto et al., 2021) and time-resolved measurements of surface height (Steen-Larsen, 2020a; Zuhr et al., 2021; Steen-Larsen et al., 2022). More details are found in Section 2.

Using these data, we demonstrate that while there is inconsistent post-depositional modification of $\delta^{18}O$ during the summers and interannually, $d$ shows more consistent modification in summer snow layers on weekly and interannual timescales (Section 3). We explore the potential mechanisms causing these signals; some behavior can be explained by existing models, but not all (Section 4). Implications of these results for IEMs and interpretations of $\delta^{18}O$ and $d$ in polar snow, firn, and ice are explored (Section 4.2).

## 2    Site Description, Data, and Methods

The data and products presented here are all derived from observations at the EastGRIP ice core site located in the Northeast Greenland National Park. In Section 2.1 we present the meteorological context of our study. In Section 2.2 and Section 2.3 we present the surface snow isotope and snow profile isotope data sets, respectively. In Section 2.3, we explain the siting, extraction, handling, and processing of the snow profiles. In Section 2.4, we discuss the age-depth model applied to the snow

profile isotope data set. In Section 2.5 we discuss nuances and caveats relevant to the interpretation of the data presented here. Table 2 contains an overview of the data used in this study.

### 2.1    Meteorology: data and context

The EastGRIP site is located on a fast moving ice stream at $75°37'47''$ N, $35°59'22''$ W at an altitude of 2708 m (m a$^{-1}$, Westhoff et al., 2022). There is a PROMICE weather station located approximately 300 m south of our study site (Fausto et al.,

2021) . The site experiences persistently-high and directionally-constant winds because its location on the ice sheet results in downslope (westerly) katabatic winds and westerly synoptic flow over the ice sheet (Putnins, 1970; Dietrich et al., 2023).

The accumulation rate was measured as approximately 134-157 mm a$^{-1}$ of liquid water equivalent (l.w.e.) from snow pit studies coincident with this work (Nakazawa et al., 2021; Komuro et al., 2021). Summertime daily accumulation was measured with stake lines during the 2016-2019 field seasons (Steen-Larsen, 2020a, b; Harris Stuart et al., 2023). The bamboo stake line

was 200-m long with 1-m spatial resolution in the 2016 field season, and 90-m long with 10-m spatial resolution for the remaining field seasons. We also determined changes in monthly mean snow height from PROMICE sonic ranger data (Fausto et al., 2021) for 2014-2019, with the annual rate of change in surface height being approximately 40 cm a$^{-1}$. The top 1-m of snow has a nearly constant density profile of approximately 337 kg m$^{-3}$, presumably constant because of the persistently high winds at EastGRIP (Schaller et al., 2016; Nakazawa et al., 2021; Komuro et al., 2021). The snow surface is spatially

heterogeneous in height, with surface features smoothing slightly throughout the summer seasons (Zuhr et al., 2021, 2023).

### 2.2    Surface snow isotopes

The top 0-1 cm snow was collected along a 1000 m path parallel to the wind in the 2016 field season, and a 100 m path for the 2017-2019 field seasons (Behrens et al., 2023a; Hörhold et al., 2023, 2022b, a). During the 2016 and 2017 field seasons, samples





**Table 2.** All data used in this study listed with units, a brief description, and data source. Uncertainties are $2\sigma$ standard deviation around the means.

| Data | Units/Res. | Description | reference |
|---|---|---|---|
| Temperature | -28.5 ± 14 $^{o}$C | PROMICE weather station, hourly frequency, 2017-2019 | Fausto et al. (2021) |
| Wind speed ($u$) | 5.26 ± 4.6 m s$^{-1}$ | data source and frequency same as above | same as above |
| Wind direction | W-SW | prevailing wind direction in all seasons, data source and frequency same as above | same as above |
| Annual accumulation (a) | 134-157 mm a$^{-1}$ | derived from from snow pits, 2009-2017 | Komuro et al. (2021) |
| Annual accumulation (b) | 145, 149 mm a$^{-1}$ | snow pits, 2009-2016 | Nakazawa et al. (2021) |
| Snow stake line 2016 | 1 m hor. res., ±1 cm vert. res. | Relative surface height measurements | Steen-Larsen (2020a) |
| Snow stake line 2017-2019 | 10 m hor. res., ±1 cm vert. res. | Relative surface height measurements | Steen-Larsen (2020b); Harris Stuart et al. (2023) |
| Surface snow, 2016-2019 | $\delta^{18}O=$ ±0.22 $^{o}/_{oo}$; $\delta D=$ ±1.6 $^{o}/_{oo}$ | Daily samples of 0-1 cm snow | Wahl et al. (2022), Section 2.2 |
| Snow profiles, 2017 | $\delta^{18}O=$ ±0.22 $^{o}/_{oo}$; $\delta D=$ ±1.6 $^{o}/_{oo}$; 1-cm res, 0-10 cm; 2-cm res, 10-100 cm | Four (4) transects, 2 May 2017 - 11 August 2017, 40 profiles | Section 2.3 |
| Snow profiles, 2018 | same as above | Five (5) transects at six locations, 12 May 2018 - 06 August 2018, 35 profiles | Section 2.3 |
| Snow profiles, 2019 | same as above | Five (5) transects, 29 May 2019 - 24 July 2019, 25 profiles | Section 2.3 |

from each site were collected and bagged individually, the measured $\delta^{18}O$ was then averaged. During the 2017 field season,
snow of equal amounts was also collected daily at the same locations then mixed into one sample bag, termed 'consolidated'
samples. It was found from this work that the mean isotopic values of the individually bagged samples were the same as the
less laboriously obtained 'consolidated' samples. Mean daily surface snow isotopic content for the summers of 2018 and 2019
were therefore determined from 'consolidated' samples.





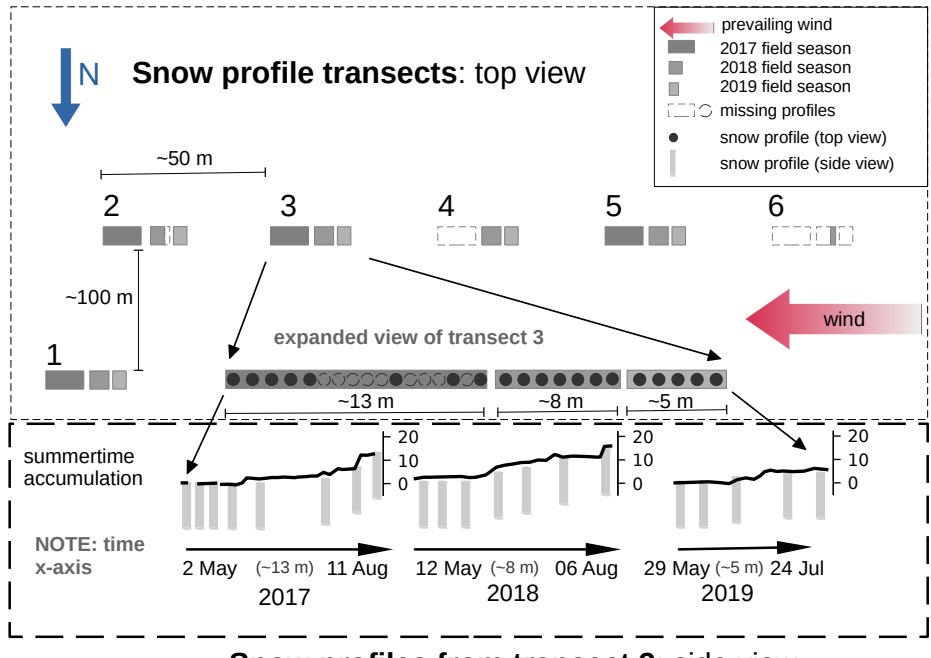

**Figure 1.** The top panel shows an overview of the relative spacing and timing of the transects along which the near-surface snow profiles were taken for this study. Each transect has the same snow profile pattern as Transect 3. The diagram is not to scale, but distances are noted. North is downward in this diagram. The prevailing wind direction is from the W-SW. The number and relative timing of snow profiles are accurately indicated. The bottom panel shows an illustration of the summertime snow stake heights along with snow profile timing. The study site is the EastGRIP ice core site in Northeast Greenland.

Once collected, either individually or as a consolidated sample, the snow was sealed in an air-tight Whirl-Pak bag and
kept frozen until measurement at the Alfred-Wegener-Institut in Bremerhaven, Gremany or the Institute of Earth Sciences in
Reykjavík, Iceland. Isotopic measurement procedures for surface snow are the same as for the snow profiles and explained in
Section 2.3.

## 2.3  Near-surface snow profile isotopes: siting, extraction, handling, and measurement

The central data presented here are isotope measurements from a time-resolved array of snow profiles from 0-1 m (Behrens
et al., 2023b). The sampling strategy is diagrammed in Figure 1. The snow profiles were taken along transects progressing in
the windward direction. Each sampling event consisted of five snow profiles taken from five unique transect lines within a few
hours of each other. The transect lines were at least 50 m from each other. A total of six transect lines were used, but only five
during any one sampling event.



The frequency of snow profile sampling events ranged between three and twenty-one days, the most common frequency
being fourteen days. Snow profiles along one transect were spaced apart by approximately one meter. The close spacing
permits us to consider most snow profiles along the same transect as representing the same snow (See Section 2.5.1). A single
snow profile was taken by gently pushing a 10-cm diameter carbon fiber tube (i.e. liner) with a 1-mm thick wall vertically into
the snow. Minimal compression of the snow column occurs during this process (maximum 2 cm, average 1 cm, Section 2.1
Schaller et al., 2016). A small pit was cleared on the downwind side of each liner so that they could be carefully extracted with
all snow stratigraphically intact. The resulting snow pit was then back-filled within two hours of the beginning of the process,
mitigating exposure of deeper layers to the atmosphere above.

After extraction, each profile was quickly transported to a cold tent for cutting and storage. The profiles were cut at 1.1-cm
resolution for the top 0-10 cm and 2.2-cm resolution for remainder of the profiles. Most profiles were not exactly 100 cm in
length due to compression and a small amount of bottom loss during extraction. The snow was cut in an open-faced tray using
a 0.10-cm thick blade. Each sample was sealed in an air-tight Whirl-Pak bag and kept frozen until measurement at either the
Alfred-Wegener-Institut in Bremerhaven, Deutschland or the Institute of Earth Sciences in Reykjavık, Island.

Measurements of $\delta^{18}O$ and $\delta D$ concentrations were made using a Picarro cavity ring-down spectrometer (models L2120-i,
L2130-i, L2140-i) and reported in per mille ($^\circ/_{oo}$) notation as shown in equation 1 on the VSMOW/SLAP scale. Memory and
drift corrections were applied using the procedure in (Van Geldern and Barth, 2012). We calculated the combined standard
uncertainty (Magnusson et al., 2017) including the long-term uncertainty and bias of our laboratory by measuring a quality
check standard in each measurement run and including the uncertainty of the certified standards. The combined $1\sigma$ uncertainty
in $\delta^{18}O$ is 0.11 $^\circ/_{oo}$ and for $\delta D$ is 0.8 $^\circ/_{oo}$ for all isotopic measurements. We focus on $\delta^{18}O$ and $d$ for the remainder of this
paper, as $\delta^{18}O$ and $\delta D$ are equivalent for our purposes.

The snow cores we use are 1-m in length to capture at least two annual cycles at EastGRIP. Modeling also indicates this
maximum depth will be well beyond the direct isotopic influence of the atmosphere (Town et al., 2008b). The spacing between
transects (approximately 50 m) is well beyond established isotopic spatial autocorrelation lengths in polar snow (Münch et al.,
2016), providing several independent realizations of the near-surface snow during each sampling event.

## 2.4 Intercomparison of chronological layers

### 2.4.1 Depth adjustment

Photogrammetric experiments at EastGRIP show that chronological layers of snow are continuous but inhomogeneous in
thickness and spatial distribution (Zuhr et al., 2021). This is in agreement with prior efforts documenting wind-driven erosion
and deposition in snow (e.g., Fisher et al., 1985; Colbeck, 1989; Filhol and Sturm, 2015). Important precipitation events will
have uneven representation in the snow, and in extreme cases (high winds with low accumulation) entire annual layers could
be scoured at polar sites with low accumulation (e.g., Epstein et al., 1965; Casado et al., 2018). Zuhr et al. (2023) confirm the
layers are continuous at EastGRIP, with only one exception in their spatial study. Zuhr et al. (2023) also documented that uneven
surfaces and concomitant heterogeneous distribution of precipitation result in spatially heterogeneous isotopic concentrations



of snow. As such, a perfectly horizontal average of $\delta^{18}O$ across the snow at any level then represents a mixture of events across time (Münch et al., 2017). Zuhr et al. (2023) estimates that the $2\sigma$ spread around mean $\delta^{18}O$ values as a function of depth is 2.9 $°\!/_{oo}$ due to the impact of this stratigraphic noise. Note, this number represents the distribution of the measurements, not the confidence in snow profile mean values.

For this study, tracking chronological layers is critical to separate wind-driven spatial heterogeneity in $\delta^{18}O$ and $d$ from other processes at work in the near-surface snow. To better align chronological layers, we apply a local depth adjustment to individual snow profiles from 2018 and 2019 based on snow stake measurements of surface height changes at each sample site, illustrated in Figure 2. For the 2017 snow profiles, we apply one depth adjustment to all profiles collected on the same day. We use the mean change in height from the 90-m snow stake transect to adjust snow surface height relative to the first profiles of the season collected on 2 May 2017 (Steen-Larsen et al., 2022). Compaction was not considered in the depth adjustment between snow profiles.

### 2.4.2 Age-depth model

The depth correction mitigates much of the stratigraphic noise induced by simple horizontal averaging, but not all. We developed an age-depth model for each individual snow profile to further minimize stratigraphic noise in chronological layer intercomparisons within one season, also allowing better intercomparison of 'same-era' snow (i.e. snow from the same time period) between profiles extracted during different field seasons.

An illustration of the age-depth model process is shown in Figure 2 and follows many similar seasonal and interannual studies (e.g., Shuman et al., 1995; Bolzan and Pohjola, 2000; Kopec et al., 2022). The end date for every profile is the extraction date, which is known precisely. From this date we worked downwards in the snow and backwards in time. Local maximum and minimum $\delta^{18}O$ values were found automatically. Dates assigned to the $\delta^{18}O$ values are from the maxima and minima in monthly mean temperature as measured at the nearby PROMICE weather station. We find at least two dates per annual layer, a summer maximum and a winter minimum.

The age-depth model depends on the continuity of layers between cores and serves to align layers to each other, as well as to the assigned dates. Although the age-depth scale is very accurate, much of our analysis depends primarily on the alignment of layers rather than the absolute date. The bottom of each snow profile was assigned dates based on contemporaneous accumulation information.

The age-depth model uncertainty comes from two sources: 1) snow profile peak and trough identification, and 2) maximum and minimum date attribution. The snow profile peak identification is more accurate near the top of the profiles because of sampling resolution. Maximum and minimum air temperatures date attribution is more accurate in summer than winter. We conservatively assess the $2\sigma$ uncertainty as a minimum of $\pm9$ days for the top of each profile, $\pm25$ days around each summer peak below 10 cm, $\pm33$ days around each winter trough below 10 cm.

A detailed discussion of the age-depth model and error analysis can be found in Section 2.5.3.



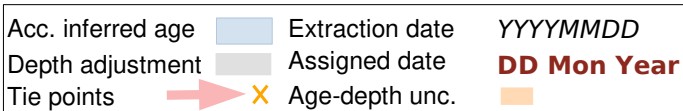

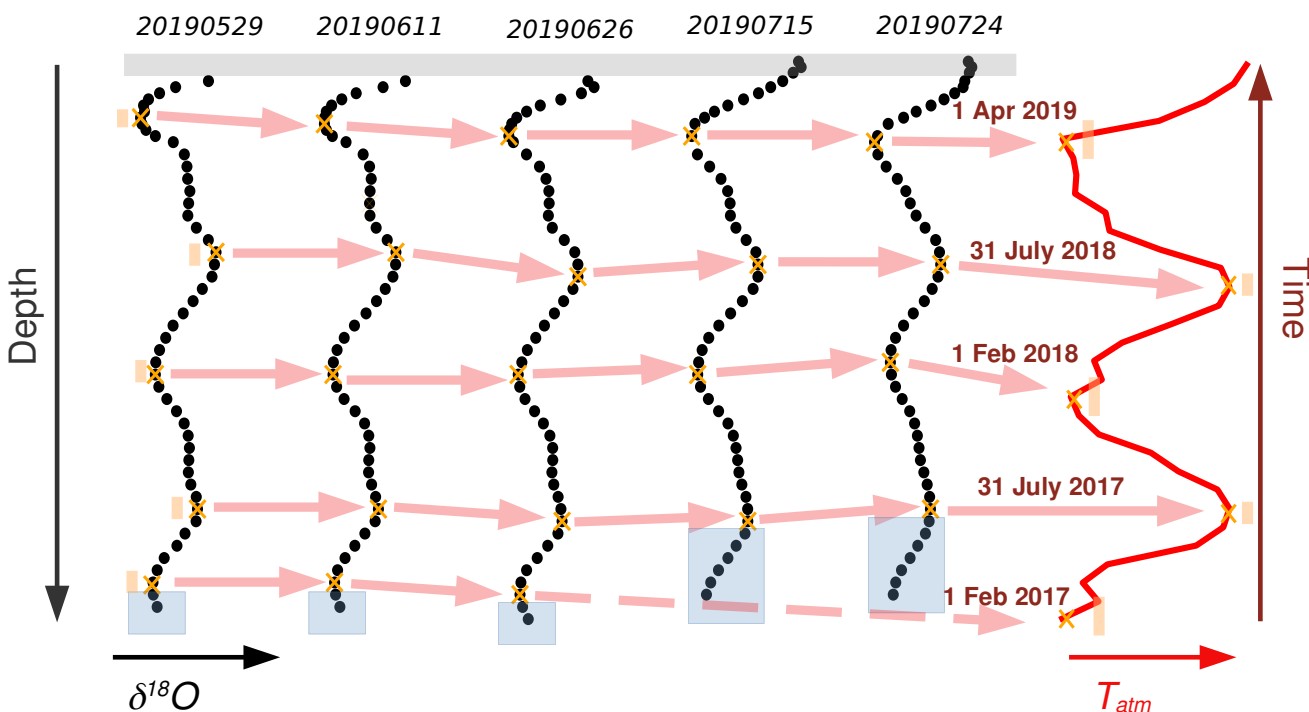

**Figure 2.** An illustration of the depth adjustment (gray range) and age-depth model applied to $\delta^{18}O$ data from transect 2 during the 2019 field season. The yellow crosses represent automatically found peaks in $\delta^{18}O$ (black dots) and monthly mean 2-m temperature (red line). Each yellow star is assigned a date, and the intervening dates are linearly interpolated to a depth value. The lowest few $\delta^{18}O$ data points are assigned by an iterative process based on the accumulation rate from that time period and then manually checked. Uncertainty in the snow profile peak identification and air temperature date assignment are illustrated in orange.





## 2.5 Snow isotope data set caveats and nuances

### 250 2.5.1 Decorrelation distances and snow profile comparisons

Our sampling and data processing strategy is designed to separate spatial and temporal variability of isotopic content of the near-surface snow. The sampling strategy is inherently destructive, which results in trade-offs between accurate sampling and monitoring temporal variability. We attempt to balance these trade-offs by sampling at approximately 1-m spacing along each transect. The 1-m spacing keeps each profile well within established spatial decorrelation distances for spatially successive
water isotope samples (1.5 m) for a polar site with a similar altitude and accumulation rate (Münch et al., 2016). Sampling closer than 1-m along a transect risks disturbing the subsequent adjacent profile.

We note that the decorrelation distances derived in Münch et al. (2016) were done without application of spatial depth adjustment or an age-depth model to align chronological layers. Their decorrelation distances then represent an overestimate for our data set after the application of depth adjustments, and an extreme overestimate after application of our age-depth
model. Zuhr et al. (2023) show that isotopic continuity of layers is the rule rather than the exception at EastGRIP.

We collected samples as close together as possible so that snow profiles taken along one transect will be considered representative of the same snow, taking into account the small amount of stratigraphic noise observed on these spatial scales (e.g., Zuhr et al., 2023). During 2017 many (18) profiles were taken along each transect although not all used here (only 8). The reason for their removal is discussed in Section A1. The higher number of snow profiles represents a larger distance traveled
over the history of the transect in 2017. It is very likely that the snow extracted from a transect at the beginning of the 2017 season does not represent the same location as the snow from the end of the 2017 season along the one transect. We consider this later when examining intraseasonal evolution of the near-surface snow.

The transect lines are separated by 50 m or more to provide 'independent' representations of the snow surface. In this way, when we average snow profiles taken on the same day, we are taking averages independent of the influence of local dune and
sastrugi features.

### 2.5.2 Mitigated biases due to sampling

We promptly back filled of each extraction sites to mitigate the influence of near-surface meteorology on the next upwind profile. High temperature gradients take days to weeks to propagate through the snow these distances (Town et al., 2008a). The potential influence of forced ventilation on near-surface snow due to tapers off dramatically after about 50 cm (Town et al.,
2008b). So, our sampling procedure sufficiently prevents unintended post-depositional change due to extra exposure to the near-surface atmosphere. Other details related to missing data and uncertainties are shared in Appendix A3.

### 2.5.3 Age-depth models

Similar age-depth models have been developed using temperature-to-isotope data sets from the Greenland ice sheet. Higher accumulation rate sites like Summit, Greenland allow more tie points in one year (e.g., Shuman et al., 1995; Bolzan and Pohjola,





2000). However, much lower resolution is also common. Kopec et al. (2022) assign 1 January to all winter $\delta^{18}O$ minima. While this is problematic for absolute dating accuracy because of the quasi-coreless winter over Greenland, it does not change their conclusions. One tie point per year works in their study because Kopec et al. (2022) are concerned with relationships between variables measured in the snow, and Summit has a fairly constant accumulation rate, albeit a seasonal compaction rate (Dibb and Fahnestock, 2004; Howat, 2022).

In our case, EastGRIP has approximately half the accumulation rate of Summit, Greenland, the accumulation rate varies with season (higher in summer and autumn, lower in winter and Spring), and the compaction rate very likely has the same seasonality as Summit. We find at least two tie points each year, which is more than sufficient to resolve the seasonal accumulation and compaction at EastGRIP. The seasonal scaling applied to the $\delta^{18}O$ and $\delta D$ time series are the same, so the position of the $d$ time series relative to $\delta^{18}O$ remains the same. This will be important when evaluating the seasonality of $d$ as the snow ages.

## 3   Results

We share results for surface and near-surface snow samples focusing on evolution during summer-only time periods (Section 3.1) separately from the interannually evolution of the snow profiles (Section 3.2). Their combined meaning are explored in Section 4.

Statistically, we are mainly concerned with how mean values compare even as distributions of these isotopic values and their derivatives may overlap. As such, most of our error values and uncertainty ranges are represented as two times the standard error around the means ($2\sigma_{\bar{x}}$, $p < 0.05$). Where the overlap of distributions are important we report two times the standard deviation around the mean (i.e., $2\sigma$).

### 3.1   summer season $\delta^{18}O$ and $d$

Figure 3 provides a look at the isotopic evolution of the near-surface snow during the summer field seasons. The extraction dates (upward arrows), 2-m air temperature, and mean surface height changes from the bamboo stake line are provided for context. The mean surface height changes do not match up exactly with the height changes in the contour plots below because they were in different locations. Each upward arrow represents the mean of four or five snow profiles taken on the same day from different transects. Aggregating the snow profiles in this manner likely mitigates much spatial variability. We show the first annual cycle (0-50 cm) because there is no detectable subseasonal change below approximately 20-30 cm on these time scales, consistent with modeling by Waddington et al. (2002) and Town et al. (2008b).

These data show that new accumulation can bring in a range of $\delta^{18}O$ values to the surface snow, but typically has a high ($\geq 10$ ‰) $d$ content. During periods of low-to-no accumulation there are coincident increases in $\delta^{18}O$ and decreases in $d$ in the near-surface snow. Figures 4-6 illustrate the changes in mean daily profiles during similar low-to-no accumulation periods. Significant increases ($p < 0.05$) in $\delta^{18}O$ are seen the summers of 2017 and 2019, down to 20-30 cm. Coincident decreases in $d$ are also seen in these difference plots, but not to $p < 0.05$. Temporal changes in the 2018 snow profiles are not so easily encapsulated in a profile difference plot shown. In this case there is no significant change in $\delta^{18}O$ and $d$ over the chosen low-





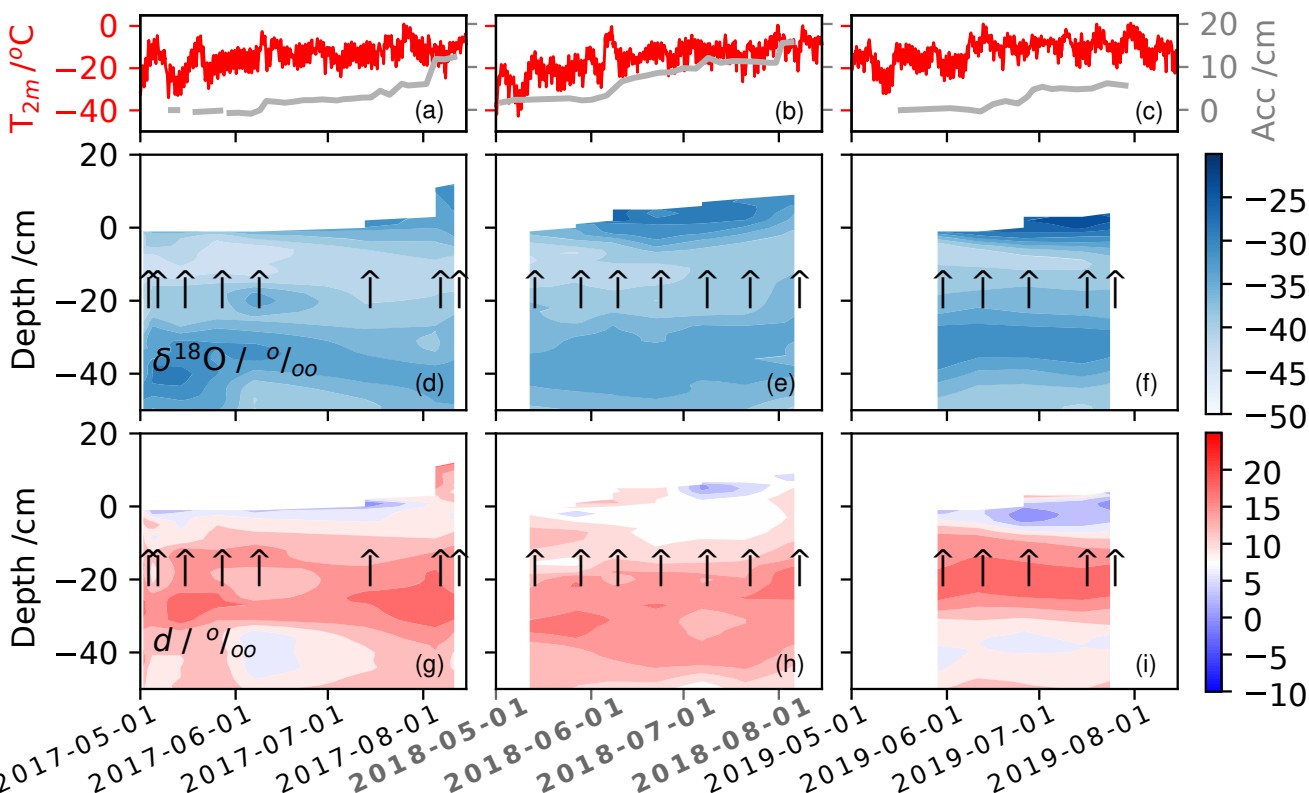

**Figure 3.** Mean $\delta^{18}O$ and $d$ snow profiles plotted as depth (vertical axis) and date of extraction (horizontal axis) for the three summer field seasons 2017, 2018, and 2019. Panels (a-c) show the 2-m air temperature from the local PROMICE weather station and accumulation from the bamboo stake line. Panels (d-f) show the $\delta^{18}O$ content of the near-surface snow determined from mean $\delta^{18}O$ snow profiles. Each arrow represents a mean snow profiles spaced approximately 50 m apart (four snow profiles in 2017, five snow profiles in 2018 and 2019). Panels (g-i) are similar contour plots but for $d$. Note the vertical axis only extends to 50 cm depth.

to-no accumulation period. Other periods during 2018 may show significant differences in their profiles, but we choose here to keep the time periods as similar as possible for this illustration.

There is a 5 $^o/_{oo}$ decrease in $d$ when comparing surface snow samples to same-era snow from a snow profile from the same summer in 2018 and 2019; the same decrease in $d$ is not apparent in the 2017. The $d$ decreases are discussed in Section 4.1.1, illustrated in Figure 8d, and shown in Table A5.



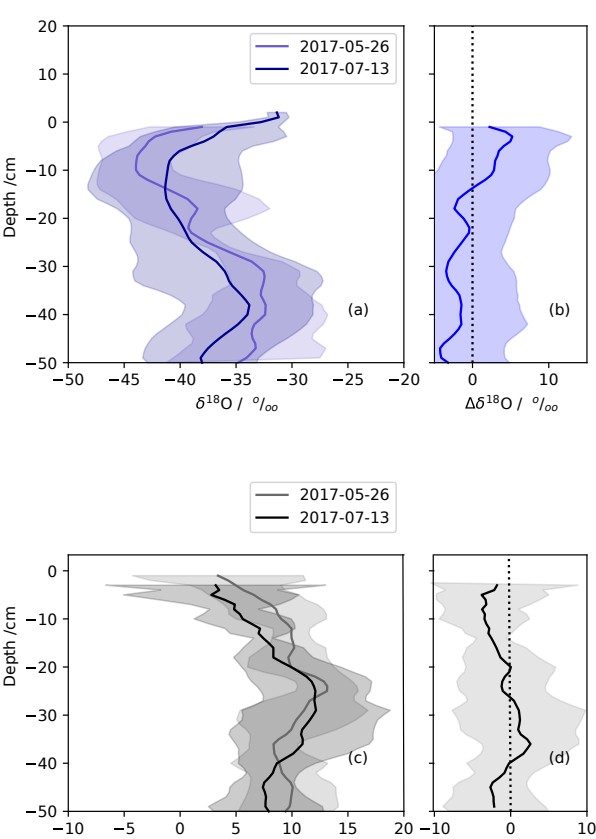

**Figure 4.** The mean isotopic change in near-surface from a low accumulation period during summer (25 May 2017 and 13 July 2017). Panels (a) and (c) are the mean snow profiles of $\delta^{18}O$ and $d$ computed from four snow profiles each. Panels (b) and (d) show the isotopic change over this time period. Error bars are $2\sigma$ standard error.

## 3.2 Interannual $\delta^{18}O$ and $d$

Figures 7 and 8 show annually successive surface and near-surface snow isotopic content for $\delta^{18}O$ and $d$, respectively. Figures 7a and 8a show the mean profiles as a function of relative depth with two standard error ($2\sigma_{\bar{x}}$) shading. The 0 m level was chosen as 29 May 2019, the day the first snow profile was extracted during in 2019. Figures 7b and 8b show the difference between each profile as a function of relative depth. These difference profiles represent the isotopic change due to aging in the firn because the same-era layers have been aligned through the depth correction, although some spatial variability no doubt remains.

Figures 7c,d and 8c,d show the same isotopic data as in their respective panels (a,b), but now plotted as a function of the age-depth model described in Section 2.4.2. The age of a given snow profile ranges from two to three years depending on the total



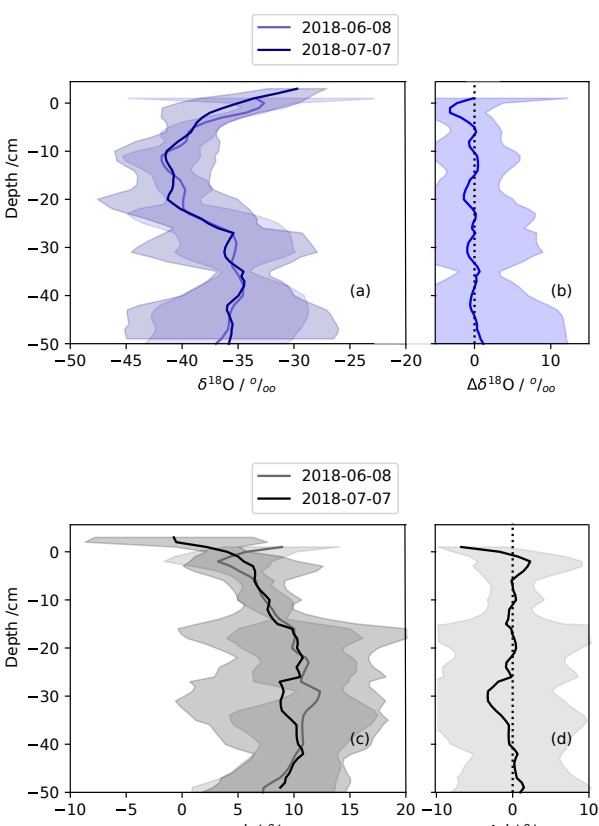

**Figure 5.** The mean isotopic change in near-surface from a low accumulation period during summer (08 June 2018 and 07 July 2018), similar to Figure 4. Panels (a) and (c) are the mean snow profiles of $\delta^{18}O$ and $d$ computed from five snow profiles each. Panels (b) and (d) show the isotopic change over this time period. Error bars are $2\sigma$ standard error.

accumulation rate for that time period and exact location. Taken as a whole, the dates represented by the snow profiles span 2014-2019. The age-depth model inherently better aligns chronological layers than the depth adjustments, further mitigating impacts of spatial inhomogeneity in stratigraphy and densification on quantitative comparison $\delta^{18}O$ and $d$ in snow layers. This can be seen in a decrease $2\sigma_{\bar{x}}$ values from panel (a) to panel (c) in Figures 7 and 8, particularly for the 2017 snow profiles.

Figure 10 shows the difference between annually successive mean snow profiles. It is similar to panel (d) from Figures 7c,d and 8 but with $2\sigma_{\bar{x}}$ shading. Figure 10 can be interpreted as how $\delta^{18}O$ and $d$ evolve one or two years after being buried, now as a function of reference snow profile age.

Annual and seasonal statistics from Figures 7 and 8 are shown in Tables A1-A4 in Appendix B.



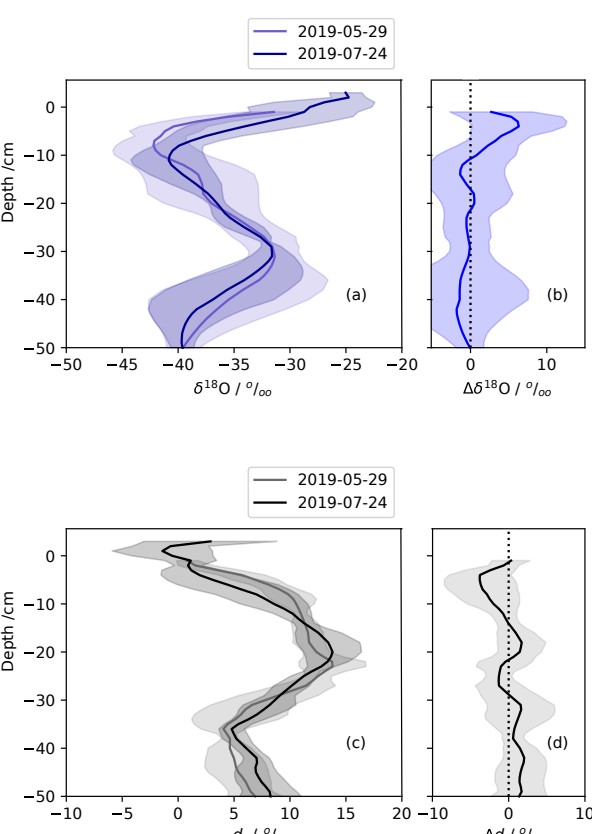

**Figure 6.** The mean isotopic change in near-surface from a low accumulation period during summer (29 May 2019 and 24 July 2019), similar to Figure 4. Panels (a) and (c) are the mean snow profiles of $\delta^{18}O$ and $d$ computed from five snow profiles each. Panels (b) and (d) show the isotopic change over this time period. Error bars are $2\sigma$ standard error.

### 3.2.1 Interannual evolution of $\delta^{18}O$

Mean annual $\delta^{18}O$ values are fairly constant throughout this time period regardless of aging, approximately -36 $^o/_{oo}$. However, there is significant variability in the peak summer $\delta^{18}O$ in each profile, regardless of snow age. The 2019 summer has the greatest peak $\delta^{18}O$ values. There is not concomitant variability in the minimum winter $\delta^{18}O$ values in this record. Some differences between profiles seem significant when plotted against relative depth (Figure 7b). However, when the age-depth model is applied, differences between profiles show no significant interannual change in $\delta^{18}O$ (Figures 7d and 10a).

During the season of extraction, the surface snow $\delta^{18}O$ values (purple squares) and mean summer snow profile $\delta^{18}O$ values match and have approximately the same variability for this period. After aging one year, the mean snow profile $\delta^{18}O$ for July




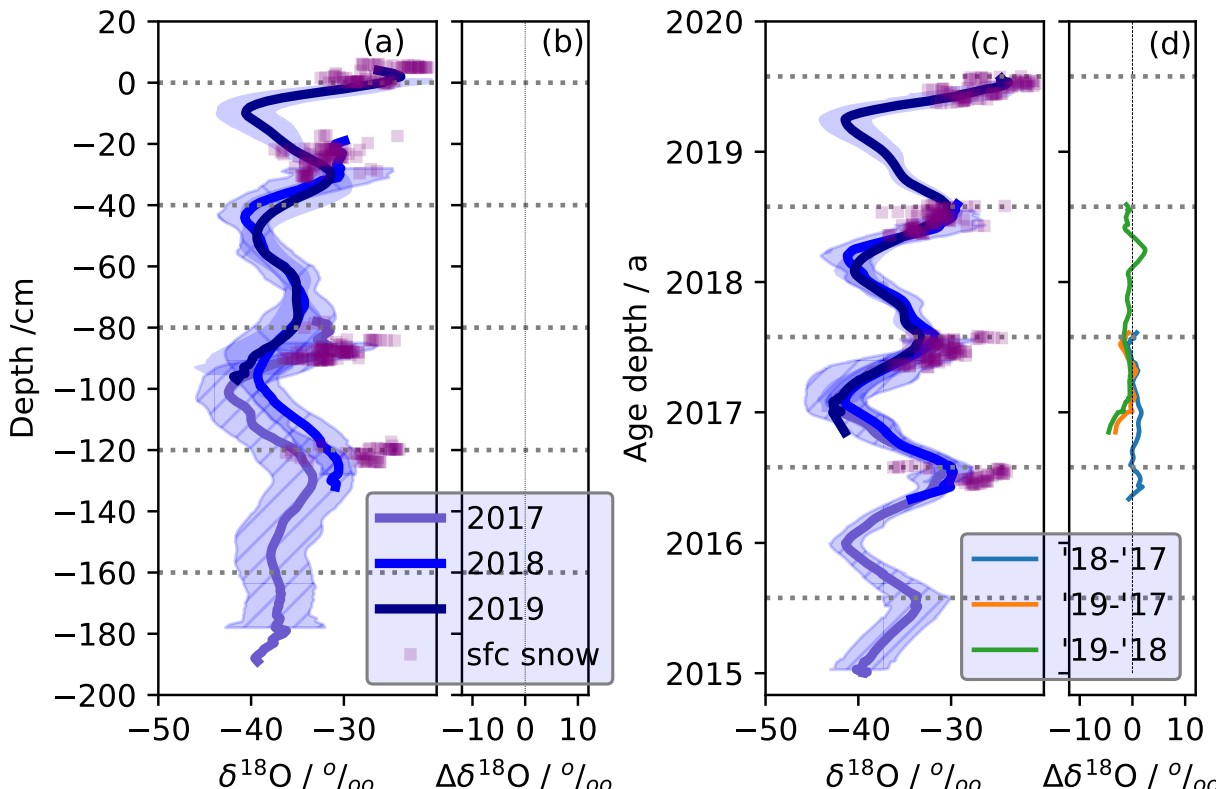

**Figure 7.** Mean $\delta^{18}O$ values from snow profiles and surface snow. The surface snow data (purple squares) are daily means from the 2016-2019 summer seasons. The snow profiles are mean values grouped by year of extraction (e.g. 2017, 2018, and 2019). Panel (a) shows the mean surface snow and snow profile $\delta^{18}O$ values as a function of relative depth. The surface is defined as 29 May 2019, the first day of snow profile sampling in 2019. Panel (b) shows the difference between each profile as a function of relative depth, representing the interannual change in $\delta^{18}O$. Panel (c) shows the mean surface snow and snow profile $\delta^{18}O$ values as a function of age-depth. Panel (d) shows the difference between each profile as a function of age-depth, representing the interannual change in $\delta^{18}O$. Shading represents $2\sigma$ standard error ($2\sigma_{\bar{x}}$). The horizontal lines in panels (a) and (b) are set at 40 cm, the approximate annual accumulation rate at EastGRIP. The horizontal lines in panels (c) and (d) represent 31 July of each year.

2018 extracted in 2019 matches the mean surface snow $\delta^{18}O$. However, the surface snow $\delta^{18}O$ from 2016 and 2017 are several $^o/_{oo}$ enriched over the snow that has aged one or two years (Figure 7c,d).





### 3.2.2   Interannual evolution of deuterium-excess ($d$)

The interannual variability and seasonal cyles of deuterium excess are shown on both depth and age-depth scales in Figure 8. The minima occur during the spring and summer while the maxima occur during autumn at the top of the profiles. This changes as the profiles age, with differences between $d$ profiles showing a distinct peak in the summer layers (Figures 8d and 10b). The surface summer snow starts with a relatively high $d$ value that decreases by as much as 5 $‰$ by the time it is extracted as a snow profile (Section 3.2.2). After aging for one year, the same summer layer $d$ has increased up to 5 $‰$

because the autumn maximum peaks broaden into summer and spring layers. Although exceeding $2\sigma_{\bar{x}}$ significance, there is also a persistent decrease in winter $d$ values shown in Figure 8d as the snow ages interannually. The mean annual $d$ values of the snow profiles do not change from year-to-year, regardless of aging (Table A3).

These profile data also demonstrate significant differences between summer $d$ values from surface snow (purple squares) and the snow profiles during the season of extraction. The mean surface snow $d$ is $10.3 \pm 2.5$ and $8.1 \pm 2.4$ $‰$, during summers

of 2018 and 2019, respectively. Whereas, the mean snow profiles show $d$ values of only $5.4 \pm 0.5$ and $3.7 \pm 0.6$ $‰$, for summers of 2018 and 2019, respectively. This represents an increase in $d$ of the surface snow of approximately 5 $‰$ in both summer season snow layers (Figure 3(h,i) or Table A5). In 2017, $d$ in the summer snow profile is less than surface snow, but the difference is insignificant.

### 4   Discussion

There are significant changes in the isotopic content of near-surface snow after deposition at the EastGRIP site. We observe these changes occurring on two timescales, during the summer season and interannually. The largest changes we observe are in the summer snow layers on both timescales. Enrichment in $\delta^{18}O$ and a decrease in $d$ can happen during the summer season in the top 20-30 cm of snow during low-to-no accumulation periods. A subsequent increase in $d$ in the summer snow layer occurs as the snow ages one or two years in the firn. Below we discuss potential mechanisms for these processes and their

implications, and make recommendations for future work.

### 4.1   Post-depositional isotopic processes at EastGRIP

It is useful remember that the phenomena of post-depositional isotopic modification is driven by latent heat fluxes to and from the *snow surface*, and latent heat fluxes within the *near-surface snow* (i.e. snow metamorphism). As primarily an observational effort, we are able to make strong inferences about potential mechanisms through compositing and context, but we also use two

models of relatively simple complexity to help constrain our inferences. The first model simulates IGD diffusion within the snow (Johnsen et al., 2000), a mechanism of snow metamorphism and an important smoothing influence of the snow on itself. The implementation of this concept is taken from the SNOWISO model (Wahl et al., 2022), and is unaltered for our use. The second model assesses the influence of atmospheric vapor deposition and snow sublimation on internal snow layers through a model forced ventilation in snow (Town et al., 2008b). The handling of isotopes in Town et al. (2008b) has been improved from



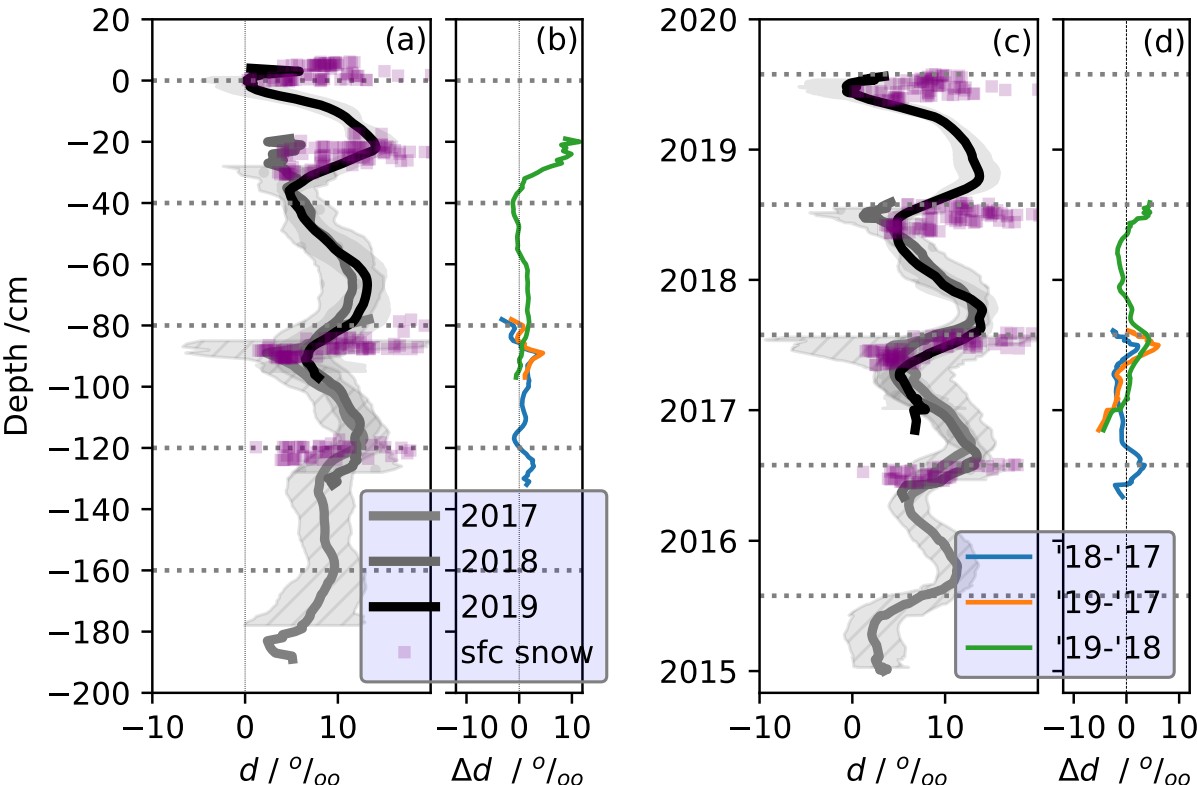

**Figure 8.** Mean $d$ values from snow profiles and surface snow, as in Figure 7 for $\delta^{18}O$. The surface snow data (purple squares) are daily means from the 2016-2019 summer seasons. The snow profiles are mean values grouped by year of extraction (e.g. 2017, 2018, and 2019) with $2\sigma_{\bar{x}}$ as the shading. Panel (a) shows the mean surface snow and snow profile $d$ values as a function of relative depth. The surface is defined as 29 May 2019, the first day of snow profile sampling in 2019. Panel (b) shows the difference between each profile as a function of relative depth. Panel (c) shows the mean surface snow and snow profile $d$ values as a function of age-depth. Panel (c) shows the difference between each profile as a function of age-depth. Panels (b) and (d) represent the change in $d$ between the different field seasons. Shading represents $2\sigma$ standard error ($2\sigma_{\bar{x}}$). The horizontal lines in panels (a) and (b) are set at 40 cm, the approximate annual accumulation rate at EastGRIP. The horizontal lines in panels (c) and (d) represent 31 July of each year.



only representing deposition of $\delta^{18}O$, to including $\delta D$ and fractionation on sublimation for both species (Lécuyer et al., 2017; Hughes et al., 2021; Wahl et al., 2022). It also has an improved equilibrium fractionation representation (Stern and Blisniuk, 2002). This model empirically combines atmosphere-to-surface latent heat flux with snow metamorphism by assuming that the atmosphere can communicate directly with subsurface snow layers.

### 4.1.1 Mechanisms at work during Summer

Section 3.1 shows evidence of a dramatic decrease in $d$ over the summer period of approximately $5\%_{oo}$, but no significant change in $\delta^{18}O$, in two summer seasons (2018 and 2019) when comparing daily surface snow samples to same-era mean snow profiles. We see mild evolution of $\delta^{18}O$ (enrichment) and $d$ (decrease) in the top 20 cm of the snow profiles as the snow ages through the summers of 2018 and 2019 (Figure 3e,f) under low-to-no accumulation periods.

Standard interpretations of a 5 $\%_{oo}$ shift in $d$ could be either a 5 K cooling of source region sea surface temperatures, or a 10 percent increase in source region relative humidity (Pfahl and Sodemann, 2014). While there is clear evidence that $d$ of precipitation does carry source region information (Dansgaard, 1964; Craig and Gordon, 1965; Merlivat and Jouzel, 1979; Jouzel and Merlivat, 1984; Pfahl and Sodemann, 2014), our results inherently impact the interpretation of $d$ in polar snow sampled after deposition.

Recent evidence demonstrates that a decrease in $d$, often associated with an increase in $\delta^{18}O$, is a likely signal of sublimation (Hughes et al., 2021; Wahl et al., 2022; Harris Stuart et al., 2023). We see mild $\Delta d$ within the near-surface snow down to 20-30 cm throughout the summer as compared to the dramatic $\Delta d$ when comparing mean $d$ from surface snow and mean $d$ from same-era snow profile. From this, we infer that much of the $\Delta d$ occurs at the snow surface before it is advected away from direct contact with the atmosphere by burial. This is consistent with observations and modeling of isotopic fractionation under summertime sublimation conditions at EastGRIP (Wahl et al., 2022). Our observations are further corroborated by a contemporaneous, high-resolution, DEM-based spatial isotope study of EastGRIP snow, completed in summer of 2019 (Zuhr et al., 2023). Using daily photogrammetry and spatially-distributed short cores (30 cm), they also observed a decrease in $d$ of up to 5 $\%_{oo}$ and no significant change in $\delta^{18}O$. The change in $d$ was concentrated in the top 5-20 cm of snow.

Challenges to the interpretation of our observations are in two categories: 1) attribution of variability temporally vs. spatially, 2) seasonally intermittency. It is clear that some variability we represent as temporal in Figure 3 can potentially be attributed to spatial heterogeneity. However, we believe this is minimal because the mean profiles are aggregated from individual profiles spaced approximately 50 m apart. So, they are immune to autocorrelation induced by wind-formed surface features, which are commonly 1-2 m in length but can have widths of up to 10 m (Filhol and Sturm, 2015). Furthermore, low-to-no accumulation periods generally exhibit increases in $\delta^{18}O$ and decrease in $d$ consistent with established 'sublimation' signals (Wahl et al., 2022). Alternatively, low-to-no accumulation periods do not show other combinations of changes in $\delta^{18}O$ and $d$.

The summertime signals are inconsistent interannually even when composited by low-to-no accumulation, emphasizing the point that post-depositional isotopic change in near-surface snow is likely induced by local meteorology. The influence of the atmosphere on surface and near-surface snow is of primary concern during low-to-no accumulation time periods, but mechanical mixing is also of concern. Wind-driven mixing complicates any interpretation, itself likely a source of local climate signal.





We have observed that toppling of surface faceting by winds (Gow, 1965) can be a significant imprinting and redistribution
mechanism at EastGRIP. Excursions of entire seasons are possible at low accumulation sites (e.g., Epstein et al., 1965). Zuhr
et al. (2023) very likely identify a missing winter layer at one of their locations, although they do not resolve an entire annual
cycle in any profile. Unraveling these processes for EastGRIP, or any site, likely requires a process-based approach to constrain
the contribution of relevant phenomena.

### 4.1.2 Modeling of summertime post-depositional processes

We explore the summertime results with three idealized scenarios. The first looks at how the isotopic gradients in EastGRIP
snow influence the signal itself through IGD diffusion (called IGD$_{summer}$). We apply this to the steepest mean isotopic gradi-
ents from our snow profiles (i.e., the top of the profiles), an extreme 'summer' scenario with snow temperatures of -11 $^oC$ for
60 days. We see an attenuation in peak $\delta^{18}O$ of up to 2 $^o/_{oo}$ due to IGD diffusion (See Figure 11). This is smaller than changes
in snow profile $\delta^{18}O$ over 47 days in 2017 and 2019 (Figures 4a,b and 6a,b), and beyond our ability to definitely discern with
420 these observations. Even so, these results point to the strong likelihood that other processes are occurring in the near-surface
snow.

The second and third scenarios investigate the potential influence of the atmosphere on the near-surface snow through forced
ventilation using a modified version of Town et al. (2008b). In these idealized scenarios, we simulate interstitial sublimation
and deposition conditions in polar snow during an idealized EastGRIP summer season. All assumptions about the model snow
properties, fractionation, and scenario conditions are summarized in Table A7, salient features are discussed here. The model
snow begins with an isotopic profile of $\delta^{18}O_{snw}$ = -30 $^o/_{oo}$ and $d_{snw}$ = 10 $^o/_{oo}$, and is allowed to change as water vapor deposits
or sublimates. The atmospheric water vapor is set to $\delta^{18}O_{atm}$ = -40 $^o/_{oo}$ and $d$ = 10 $^o/_{oo}$. The atmospheric value is constant
throughout the simulations, assuming the boundary layer drives isotopic content as it does in relatively windy places like
NEEM (Steen-Larsen et al., 2014). We assume ice saturation at all temperatures. All fractionation is considered to occur in
equilibrium. We set the snow surface structure with undulations of 1-m peak-to-peak length of and 0.10 m half-height; the
feature sizes were chosen to represent summer snow conditions at EastGRIP (Zuhr et al., 2023) and be compatible with the
model parameterizations (Colbeck, 1997; Waddington et al., 2002; Town et al., 2008b).

The deposition scenario, FV$_{dep}$, is driven a prescribed temperature difference between the surface atmosphere (T$_{atm}$ = -
10 $^oC$, a typical mean July air temperature for EastGRIP) and near-surface snow (T$_{snw}$ = -15 $^oC$). This can be considered
early EastGRIP summertime conditions when the air temperature is consistently warmer than the snow. The assumption of ice
saturation dictates that warmer saturated air deposits excess vapor in pore spaces as it is forced in through surface features,
thereby modifying the $\delta^{18}O$ and $d$ signals. The sublimation scenario, FV$_{sub}$, has the reverse conditions. This can be considered
late summertime EastGRIP conditions when air temperature is consistently colder than the snow. Under these conditions,
colder, saturated surface air warms as it enters the snow, sublimating interstitial mass as the pore spaces achieve saturation.

Figure 9 shows how the model predicts vapor exchange with the atmosphere will change with surface winds of 5 m s$^{-1}$,
the mean annual and mean summertime wind speed for EastGRIP. The model predicts the impact is largest at the surface and
tapers to insignificant levels by 20-30 cm depth. Both FV$_{dep}$ and FV$_{sub}$ predict an increase in $\delta^{18}O$ and a decrease in $d$. The



absolute magnitude of the modeled changes depend on the amount of time spent under these conditions. The relative magnitude of the change is the primary indication which process was at work, deposition or sublimation. Deposition will cause decreases
in $\delta^{18}O$ if the water vapor isotopic content is low enough.

These scenarios are idealized and only intended to help constrain interpretation of our results. The magnitude, direction, and depth of the modeled changes are consistent with changes observed in snow profile during low-to-no accumulation periods in at EastGRIP during summer (Figure 3). One can not rule out that the atmosphere has a significant influence on the near-surface snow during relatively warm summer months. Ratios of $\Delta\delta^{18}O{:}\Delta d$ are an order of magnitude greater for $FV_{sub}$ than $FV_{dep}$ in
these scenarios. A parameter like $\Delta\delta^{18}O{:}\Delta d$ may help climate interpretation of paleorecords when post-depositional isotopic change is suspected by helping assess potential impacts of post-depositional latent heat fluxes on the isotope record.

From a meteorological point of view, an important nuance is the combined choice of wind speed and length of simulation. We provide several combinations. A typical sustained wind speed at EastGRIP is 5 m s$^{-1}$ due to its location, slope, and elevation in NE Greenland. Higher wind speeds occur for much shorter durations, which we also simulate (Figures A1 and A2). The
shorter but more intense events have similar impacts isotopically on the near-surface snow. However, higher wind speeds do have the theoretical ability to reach deeper into the snow, as occurs in our model.

The assumptions of temperature in these scenarios are meant to represent the steady warming and cooling of the summertime snow. The same scenarios can also represent typical maximum and minimum diurnal temperatures. Larger variations occur on this time scale, but the temperature gradients penetrate less deeply into the snow on diurnal time scales. The impact of these
processes drops dramatically with temperature during the other seasons.

Other vulnerabilities in this modeling approach exist. Although the snow structure is based on Zuhr et al. (2023), it idealized. Zuhr et al. (2023) and others (e.g., Gow, 1965) show that surface relief on high altitude ice sheets decreases throughout the summer. More complicated representations are possible, but likely with marginal returns. We simulate the mean impact of force ventilation, but the physical phenomenon of ventilation of the snow will vary spatially under the variable surface structures.
Dunes and sastrugi migrate under blowing snow conditions (Filhol and Sturm, 2015), but this is not a pronounced effect in summer when the snow surface tends to solidify and flatten. Thus, the processes modeled here may represent an additional source of isotopic heterogeneity in addition to the heterogeneous filling observed at EastGRIP (Zuhr et al., 2021, 2023).

The model assumes direct exchange of air between the atmosphere and each snow depth. This does not happen; very likely vapor exchange is layer-to-layer through the snow. The time constants that underpin the model of surface-to-subsurface vapor
exchange (Waddington et al., 2002) and the assumption of equilibrium fractionation accommodates this weakness of the model. Kinetic fractionation does occur in the snow, but it is more likely driven by vapor-pressure gradients that derive from interstitial temperature gradients than net vapor exchange with the atmosphere.

Laboratory experiments of isotopic evolution of snow under high flow rate forced ventilation (2 L·min$^{-1}$) show isotopic changes similar in magnitude as observed and modeled here. However, the changes only extend to layers of thickness up to 3
cm (Hughes et al., 2021). Further study should be done to unify these approaches to understanding the potential impact of the near-surface atmosphere on near-surface snow.



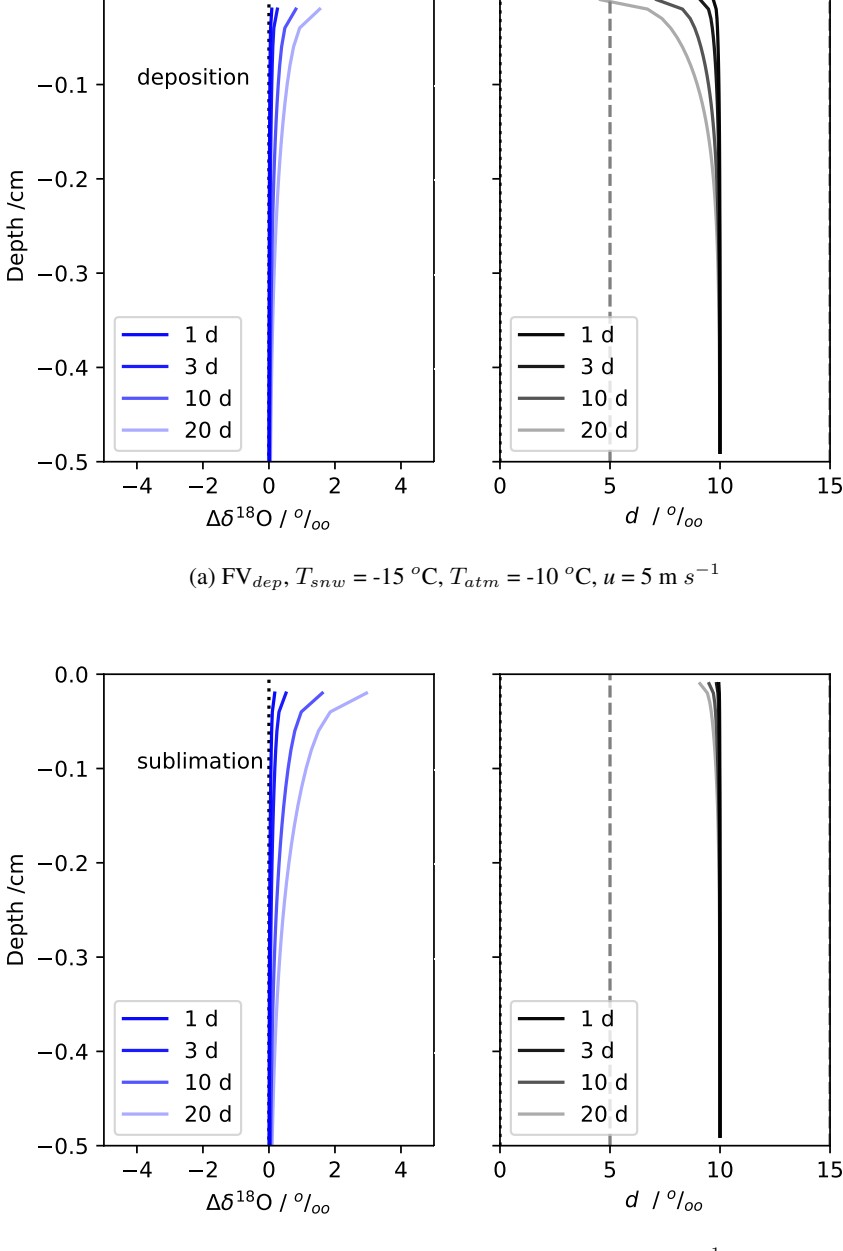

(a) FV$_{dep}$, $T_{snw}$ = -15 $^o$C, $T_{atm}$ = -10 $^o$C, $u$ = 5 m $s^{-1}$

(b) FV$_{sub}$, $T_{snw}$ = -10 $^o$C, $T_{atm}$ = -15 $^o$C, $u$ = 5 m $s^{-1}$

**Figure 9.** Idealized simulations of forced ventilation on isotropic snow under typical EastGRIP summertime conditions. The snow structure and isotopic content of snow and vapor are detailed in Table A7.



### 4.1.3 Mechanisms at work interannually

In our interannual analysis, inferences about the influence of the summertime atmosphere on the near-surface snow are strongest because the snow profiles were extracted during the summer season. However, inferences about the influence of other seasons are possible as the snow profiles typically extend approximately two and a half years.

### 4.1.4 Interannual $\Delta\delta^{18}O$

There are no significant changes in $\delta^{18}O$ ($\Delta\delta^{18}O$) between the mean snow profiles extracted during different summers (Figure 7c,d and Figure 10a). Figure 10a shows similar information to Figures 7d, but here the $\Delta\delta^{18}O$ is a function of the reference year age-depth. In other words, it shows $\Delta\delta^{18}O$ after one and two years of aging for the entire snow profile data set.

Exploring further potential changes in $\delta^{18}O$, we compute a temporally-based (seasonal) temperature sensitivity ($\gamma_t$) by taking the ratio of the difference between maximum (summer) and the minimum (winter) $\delta^{18}O$ values to the corresponding minimum and maximum monthly mean temperatures. For this, we use the same tie points as those used in the age-depth model (e.g., Figure 2). This process is based on subseasonal temperature sensitivities found in Greenland (e.g., Shuman et al., 1995; Bolzan and Pohjola, 2000) and the Antarctic (e.g., Casado et al., 2018). The $\gamma_t$ for each half year is the ratio of seasonal change (summer-to-winter, winter-to-summer) in $\delta^{18}O$ over the seasonal change in monthly mean temperature (Figure B1). We find a mean $\gamma_t$ that starts at approximately $0.297\pm0.03‰\,^oC^{-1}$.

The initial $\gamma_t$ we observe at EastGRIP is slightly lower than modern-day $\gamma_t$ values derived from microwave surface temperature retrievals or high frequency borehole thermometry for the modern day at Summit, Greenland ($\gamma_t$= 0.46 $‰\,^oC^{-1}$, $\gamma_t$= 0.54 $‰\,^oC^{-1}$, $\gamma_t$= 0.46 $‰\,^oC^{-1}$, respectively Shuman et al., 1995; Bolzan and Pohjola, 2000; Cuffey et al., 1995). Although $\gamma_t$s are considered to have more climatological fidelity than $\gamma_s$s (e.g., Cuffey et al., 1995) for reconstructing past climate from water isotope records, non-linear reconstruction methods are better at accounting for climate-related variability due to moisture source $\delta^{18}O/d$ content, transport pathway, ice sheet elevation, and local cloud conditions (Cuffey et al., 2016; Markle and Steig, 2022). These nonlinear factors are likely important to the differences observed between Summit, Greenland and EastGRIP.

The observed $\gamma_t$ at EastGRIP decreases at a rate of $0.096 \pm 0.04‰\,^oC^{-1}\,a^{-1}$, which corresponds to a 2.8 $‰\,a^{-1}$ decrease in $\delta^{18}O$ annual cycle (Figure B1). We have chosen to fit a linear pattern to account for errors in both variables (Trappitsch et al., 2018) to the decrease in $\gamma_t$, but one could argue for a more dramatic drop in $\gamma_t$ over the first 0.5 years then a much slower change in $\gamma_t$ thereafter. Until more is known about the processes at work, the assumption of linearity is the most viable null hypothesis. We model that $\Delta\delta^{18}O\,\Delta T^{-1}$ changes at a linear rate of $0.16 \pm 0.03\,‰\,^oC^{-1}\,a^{-1}$ ($p < 0.05$), using the same IGD diffusion scenarios as Section 4.1.1, which is effectively the same rate observed in the snow profiles.

Shuman et al. (1995) observed a decrease in $\delta^{18}O$ annual cycle of 1.3 $‰\,a^{-1}$ over a three-year time span, which they attribute to 'diffusion.' Summit, Greenland has similar elevation and climate to EastGRIP. The difference in annual cycle $\Delta\delta^{18}O$ can likely be explained by the higher accumulation rate at Summit ($b$ = 25 cm $a^{-1}$ l.w.e., Dibb and Fahnestock, 2004; Howat, 2022). Accumulation slows interstitial diffusion (Johnsen et al., 2000) and mitigates the influence of the atmosphere on near-surface snow (Town et al., 2008b). Kopec et al. (2022) find little to no change in isotopes between precipitation and





near-surface snow after deposition. Other processes are likely important in the surface and near-surface snow, as we infer later through examination of the $d$ signal.

     The dramatic changes in $\gamma_t$ we observe illustrate why it is difficult to use seasonal isotope-to-temperature sensitivities to reconstruct past climate. In this case, we are able to explain the increase in sensitivity solely with IGD diffusion. Our simulation does not account for other processes like temperature-gradient-driven (TGD) diffusion or interstitial heat and vapor

transport due to force ventilation. TGD diffusion likely acts to smooth isotopic signals. It alternates direction in the top 20-30 cm synoptically, and diurnally during sunlit periods. Seasonally, the TGD diffusion points in one primary direction below 20-30 cm. Forced ventilation likely acts to bias isotopic content of the snow based on the isotopic content of the overlying atmosphere. Proper reconstruction of climate variables from water isotopes requires explicit consideration of these processes to avoid misattribution or over-attribution of processes to observed changes.

**4.1.5   Accumulation rate from depth and $\delta^{18}O$**

The evolution of the $\delta^{18}O$ annual cycle is small enough for the $\delta^{18}O$ signal to be put to other uses. Using the summer $\delta^{18}O$ profile peaks as annual markers, we find a mean annual accumulation rate across all snow profiles of 45.6 ± 3.8 cm (13.5 ± 1.1 cm a$^{-1}$ l.w.e.) for this time period (2014-2019). This is consistent with accumulation rates for EastGRIP just prior to the observation period derived using a similar method (Nakazawa et al., 2021; Komuro et al., 2021), as well as coincident estimates

of surface height change from PROMICE sonic rangers (Fausto et al., 2021) when a compaction rate for EastGRIP is applied (Macferrin et al., 2022).

     The annual surface height change is fairly continuous at EastGRIP (Fausto et al., 2021), more snow comes in the summer and autumn over winter and spring. This seasonal accumulation weighting explains the differences between the residual profiles in Figures 7b and 8b, and 7d and 8d.

**4.1.6   Interannual $\Delta d$**

Figures 8c,d and 10b show a significant increase in $d$ in summer layers, but no significant change in other seasonal layers. The consistent pattern evident here is a 3-5 ‰ increase in $d$ after one year of aging that sustains into the second year. Reexamining these results through $\delta D$-to-$\delta^{18}O$ relationships, the summer layers during their first year in the snow have a slope of 7.87 ‰· ‰$^{-1}$, which changes to 8.56 ‰· ‰$^{-1}$ after one year in the snow (Table A6). This represents a dramatic resetting of the

meteoric water line relationship.

     Modeled changes in $d$ due to IGD diffusion shows some important similarities to the observations (Figure 11). IGD diffusion naturally causes the most dramatic changes around the highest gradients. The model predicts a large $\Delta d$ on the order of +5 ‰ after one year of diffusion in the top spring snow layer, which steadily decreases in subsequent years. We see the same initial increase after one year, but the continuing positive $\Delta d$ does not occur in the following year. Furthermore, because this is IGD

diffusion smoothing, each positive $\Delta d$ predicted by the model is associated with a negative $\Delta d$ of very similar magnitude; the negative $\Delta d$ excursions are not observed in the snow profile changes shown in Figure 10.





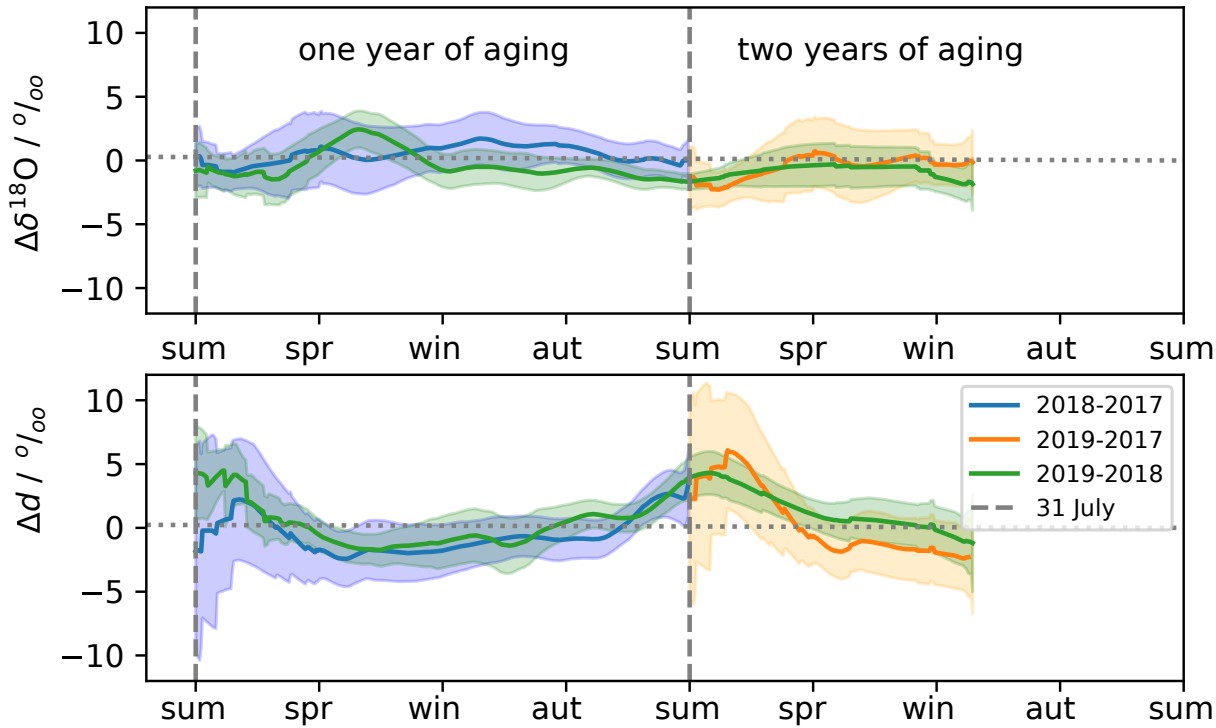

**Figure 10.** The change in $\delta^{18}O$ (panel (a)) and $d$ (panel (b)) after one-to-two years of aging in the near-surface snow. The change is determined as the difference between profiles shown in Figures 7c and 8c, and plotted as a function of the age of the reference profile in the difference. Seasons are marked on the horizontal axis, with snow depth increasing and time decreasing to the right.

The mechanisms at work interannually are then likely a combination of IGD diffusion and other post-depositional processes. It seems logical that the large reservoir of the atmosphere because we are looking for processes that bias the isotopic signal towards an increase in $d$. Our simulations indicate that the near-surface atmosphere likely causes a negative $\Delta d$ during summer

through surface sublimation but possibly through force ventilation of near-surface snow. We see from sonic rangers that two-thirds of snow height changes occur in the short summer and autumn. So, it is probable that the snow has been sufficiently advected away from the influence of the atmosphere by mid-to-late autumn, mitigating the influence of the atmosphere on the snow after this time.

Thus, another interstitial process is likely involved. Temperature-gradient-driven diffusion is a viable candidate but beyond

the scope of this work. A more mechanistic study is necessary to resolve specific processes and how they manifest in the



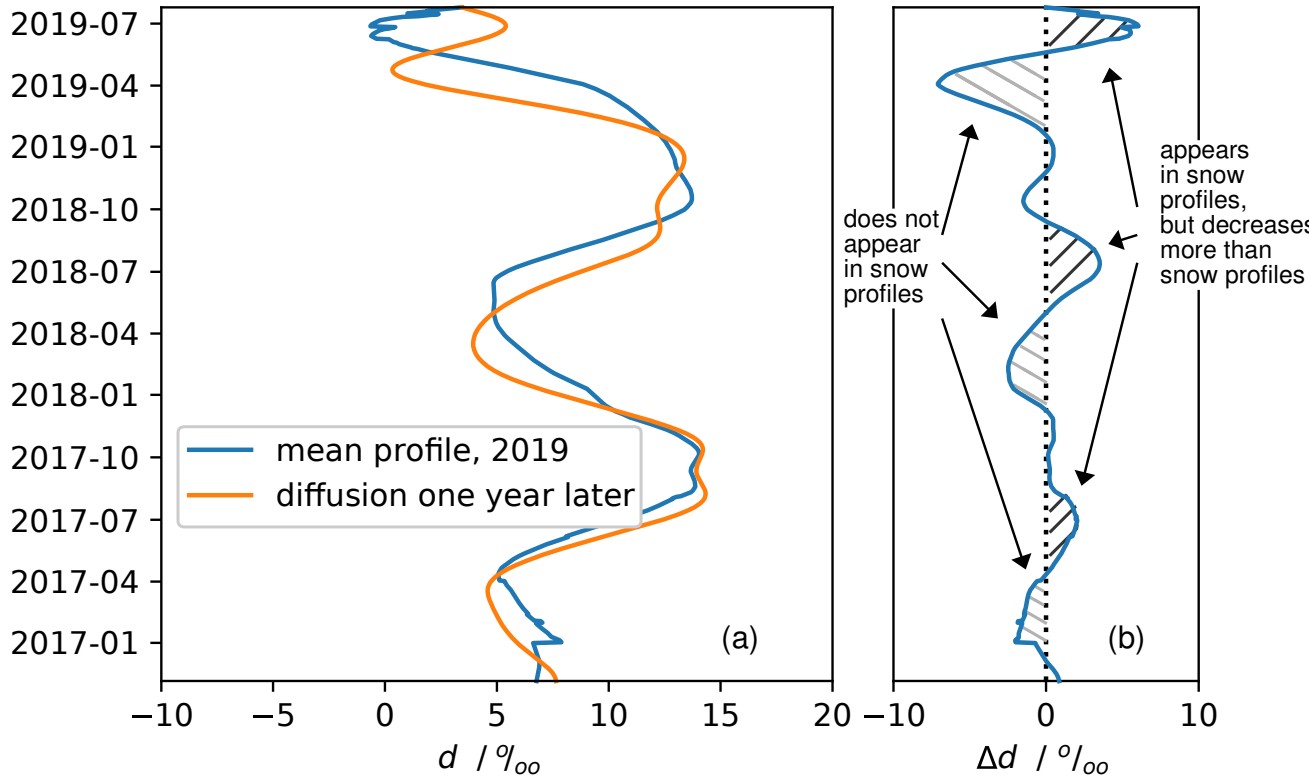

**Figure 11.** Annual isotope-gradient-driven diffusion scenario. Panel (a) shows a simulation of the impact of isotopic-gradient-driven diffusion on the mean $d$ snow profile from 2019 (blue curve) after one year of aging (orange curve). Panel (b) shows the change in $d$ ($\Delta d$) after one year of aging. The simulation is described in detail in Section C1.

context of observed meteorology. In the following section, we will explore potential avenues of research in this direction in addition to assessing implications of the results and analysis presented above.

### 4.2 Implications and Future Work

A primary aim of our study, and others like it, is to improve the understanding of how climate is recorded in polar snow. The
processes that contribute to how climate signals are recorded in polar precipitation and snow, including those outlined here, are distinct enough that conceptually separating the contributions of processes to the isotope-climate signal seems appropriate. We suggest is to dividing the contribution of processes into the following categories: *atmosphere*, *snow surface*, *near-surface snow*, and *deep firn*. The historical approach has been to consider the *atmosphere* as the source of the climate signal, and the *deep firn* as a source of noise to be inverted (e.g., IGD diffusion, advection, thinning). As stated earlier, the *atmosphere* is



often represented by climate-to-isotope sensitivities that reduce to temperature-to-isotope sensitivities, often reducing further to empirical linear relationships between spatial or temporal temperature data sets. A range of physically-nuanced models are used for dealing with the *atmosphere* category more explicitly (Werner et al., 2011; Dee et al., 2015; Hu et al., 2022; Markle and Steig, 2022), but this does not change the overall approach.

However, the mounting evidence from this work and references herein shows that some climate signal is being set locally

at the *snow surface* and in the *near-surface snow* before the snow is advected away from the influence of the atmosphere. We have observed significant post-depositional changes in surface and near-surface snow isotopic content likely due to vapor transport on two time scales, during the summer season and interannually. The post-depositional changes in $\delta^{18}O$ and $d$ during low-to-no accumulation periods in summer vary from year to year. The $d$ content undergoes a significant and likely reliable post-depositional increase in summer snow layers in one year within the firn.

This combination of evidence is particularly impactful to the interpretation of $d$. While $d$ is clearly representative of source region characteristics (e.g., Jouzel and Merlivat, 1984; Pfahl and Sodemann, 2014; Markle and Steig, 2022), we show here there are both *snow surface* and *near snow surface* components to the $d$ signal that are as much as 25 percent of the annual signal at EastGRIP. The intermittency of the summer post-depositional $\Delta d$ coupled with the reliability of the interannual post-depositional $\Delta d$ means that these changes are likely site-based and depend in meteorology. As such, we argue for a much more

process-oriented look at how water isotopes change just after deposition but while still in under the influence of the near-surface atmosphere, particularly $d$.

If the *surface snow* and *near-surface snow* represent components of the climate signal, then they deserve continued experimental and observational attention, including direct characterization of the evolution of snow properties and associated isotopic signals. Such experiments should contend with difficult snow metamorphism problems like the combined influences of surface

frost formation and snow redistribution. Flattening of surface features during summer at EastGRIP (Zuhr et al., 2021) likely has a sublimation/deposition isotopic signal related to frost formation and toppling (Gow, 1965). Where does this vapor come from, above or below the *snow surface*? How does the flattening process affect $\delta^{18}O$ and $d$? Several questions remain on how to connect snow metamorphism to the evolving isotopic content of snow.

The implications of our study also extend to the inner workings of many IEMs. The cloud phase and saturation parametriza-

tions that govern much of the isotopic signal produced by the models (e.g., Petit et al., 1991; Ciais and Jouzel, 1994; Blossey et al., 2010; Dütsch et al., 2019; Markle and Steig, 2022) are based on $d$ data from snow and ice cores (e.g., Johnsen and White, 1989; Petit et al., 1991; Masson-Delmotte et al., 2008) without considering any post-depositional change. The supersaturation parametrization is one of the most impactful, tuneable parameters in IEMs today (Dütsch et al., 2019; Markle and Steig, 2022), and at a minimum deserves representative ground truth. Similarly, a recent definition of $d$ optimized for cold climates used sur-

face snow as ground truth without assessment of the surface snow's $d$ vulnerability due to post-depositional change (Uemura et al., 2012), yet is still widely used.

If parametrizations of cloud phase and saturation can be trusted then it is appropriate to separate the weather and climate processes contributing to the *atmosphere* category of water isotope signals into: precipitation, source region isotopic content, source and 'cloud' temperatures, and regional temperature gradients (e.g., Markle and Steig, 2022). However, at the *surface*





*snow* it is clear from our evidence and others (Casado et al., 2021; Wahl et al., 2022; Dietrich et al., 2023) that water isotopes in polar snow are also directly impacted by latent heat transfer. So, this category may require additional local proxies related to the surface energy budget. Similarly, our work indicates the isotopic content of *near-surface snow* is influenced by snow temperature, snow temperature-gradients, surface wind speeds, snow structure, and accumulation rates. The *deep firn* is typically considered an invertible, 'unbiased' modifier of water isotope signals, although growing awareness of surface melt events in

records (Westhoff et al., 2022) and their potential impact on deeper layers is becoming a concern (Harper et al., 2023).

    This contribution-oriented interpretation of the $\delta^{18}O$, or $d$, proxy is particularly important to studies like Jones et al. (2023), who interpret summer-only $\delta D$ changes in West Antarctica as changes in summer temperature due to changes in insolation. Interpreting changes in $\delta D$ as both changes in temperature and latent heat flux could help explain why the West Antarctic summer $\delta D$ pattern is correlated with Milankovitch insolation patterns even though annually coincident winter correlation

in $\delta D$ is not clearly evident. Similarly, studies using $\delta^{18}O$ as summer or annual temperature proxies in ice sheet elevation reconstructions may be biased warm due to influence of sublimation on $\delta^{18}O$ (e.g., Grootes and Stuiver, 1987; Lecavalier et al., 2013; Badgeley et al., 2022), likely yielding thinner ice sheets than were actually present.

    Our proposed view of climate proxies provokes suggestions for improved field experimentation and modeling. In addition to characterizing the surface energy budgets and subsurface vapor exchange along with isotope records, it will be important

to characterize seasonally-dependent post-depositional change. Our data set primarily explains about the changes in summer snow layers during summertime and interannually. The seasonality of accumulation, temperature, and humidity are part of our detection bias, which year-round sampling would mitigate. Year-round sampling would also provide ground truth for modeling studies that explicitly include *atmosphere*, *surface snow*, and *near-surface snow* processes.

## 5   Conclusions

Water isotopes in polar snow have historically been used to infer information about past climates of polar ice sheets, as well as the integrated history of polar precipitation. These inferences rely on a contiguous physical understanding of the water's history, from source to extraction. Weak links in this understanding exist at the snow surface and in the near-surface polar snow where dynamic snow metamorphism occurs under the influence of local meteorology and climate. We provide a data set from the EastGRIP site in NE Greenland with successive views of the same snow layers, documenting how the surface

and near-surface snow age isotopically on two time scales, during summer and interannually. Our data were extracted during summer months of 2017-2019, so our conclusions about the summer layers are strongest.

    Surface and near-surface snow collected during the same summer season show isotopic post-depositional change during low-to-no accumulation events. Much of the changes we observed are in the $d$ signal. Dramatic change seems to occur right at the snow surface, before even a few weeks pass. These observations are consistent with prior studies at the same site (Wahl

et al., 2021, 2022; Zuhr et al., 2023). The combined $\delta^{18}O$ and $d$ signature is inconsistent from season-to-season, pointing to the need for more process-based understanding. The mean summer surface snow $d$ is always greater than the mean $d$ from snow profiles extracted later in the season, which is a strong indication of sublimation influences during the summer season



and consistent with summertime latent heat flux studies at EastGRIP (Wahl et al., 2021; Dietrich et al., 2023). However, it is possible that similar changes are occurring shortly after deposition in other seasons at relatively warm or low accumulation sites, particularly during early autumn and late Spring. Observations and forced ventilation modeling indicate that summer post-depositional changes can occur down to 20-30 cm in the snow. The forced ventilation model also shows that deposition can cause enrichment of $\delta^{18}O$ or decreases in $d$ through deposition if the relative isotopic content of the atmosphere is high enough.

We see significant increases in $d$ of up to 5 $\,^o\!/_{oo}$ in the summer layers after one year of aging in the firn. We see decreases in $d$ in other seasonal layers, but these decreases do not rise to the significance level of $p < 0.05$. The increases we observe in summer layer $d$ are coincidentally almost the same the magnitude as the decrease in $d$ observed during some summer seasons immediately after deposition. A significant decrease in seasonal $\Delta\delta^{18}O \; \Delta T^{-1}$ is observed in our study period, likely due to isotope-gradient-driven diffusion.

Mechanisms for the changes during summer months likely include a combination of isotopic-gradient-driven diffusion and forced ventilation of the near-surface snow. We postulate that some summer wind-driven redistribution events can have distinct sublimation/deposition signatures after surface faceting events. Interannually, isotopic-gradient diffusion can explain the changes in seasonal isotope-temperature sensitivity. It also explains some but not all of the $\Delta d$ pattern we observe. We suspect seasonally- and synoptically-induced vapor-pressure gradients in the near-surface snow to be an important metamorphic process during spring, summer, and autumn; they should be less important during winter months due to the low interstitial vapor pressures.

Our observations are relevant for the interpretation of water isotopes as proxies for past climates in polar regions. Intermittent summer enrichment of surface and near-surface $\delta^{18}O$ and $d$ during low-to-no accumulation events indicate that this proxy should likely be interpreted as a multifaceted proxy that separately represents the *atmosphere* (e.g. integrated regional and local cloud temperatures), *snow surface* (latent heat flux), and *near-surface snow* (snow properties, $T_{snw}$, $dT \; dz^{-1}$).

Our results are complicated by the extractive nature of the observations, where spatial variability is at risk of being interpreted as temporal variability. Our strategic spatially-distributed sampling program coupled with the depth corrections and an age-depth model accounts for most of the stratigraphic noise in our error bars, but of course not all.

These results are specific to the present day climate at EastGRIP, a relatively warm but low accumulation site on the Greenland Ice Sheet. These results are consistent with prior work exploring and documenting post-depositional processes (e.g. Waddington et al., 2002; Neumann et al., 2005; Town et al., 2008b; Steen-Larsen et al., 2014; Casado et al., 2021; Hughes et al., 2021; Wahl et al., 2021, 2022; Zuhr et al., 2023) and demonstrate that more general revisions to interpretations of water isotope proxies in polar snow are needed. Questions remain about potential changes in other seasons, mechanisms at work in the near-surface snow, and their relative importance. We also still lack generalized tools for assessing near-surface post-depositional modification of water isotope proxies at ice core sites, which are critical for interpretation of water-isotope-based climate records.

We recommend further field work and modeling of the annual evolution and spatial variability of the near-surface snow. Improved field data will act as a training ground for the development of process-based, isotope-enabled models that connect



the *atmosphere*, *snow surface*, and *near-surface snow*. Driving, or coupling, the near-surface snow models with meteorological IEMs will greatly advance site-agnostic, process-categorized interpretation of past climates using polar snow.

*Data availability.* Snow profile data are freely available at https://doi.org/10.1594/PANGAEA.958540. The full citation is in the reference section (Town et al., 2023).

## Appendix A: Ancillary information about snow profiles

### A1 Extra 30-cm cores taken in 2017

Figure 1 indicates that some profiles from 2017 were not used. These profiles were 30-cm in depth, and so do not contain a
full annual cycle of accumulation at EastGRIP. They are statistically equivalent to the top 30 cm of the 1-m cores. They were removed so as not to over-weight the top 30-cm of our analysis.

### A2 Age-depth model determination and uncertainty

The age-depth model was determined based on presumed correlations between air temperature and isotopic content of snow. This study challenges some basis of that assumption, but by and large we see the same patterns in mean monthly air temperature
as the $\delta^{18}O$ snow profiles. In assigning dates to $\delta^{18}O$-values, we take into account evidence-based shifts in $\delta^{18}O$ during summer due to sublimation, as well as the uncertainty induced in assigning winter dates when minimum temperatures may not be the same as minimum temperatures during precipitation events.

The date assigned summer maximum $\delta^{18}O$ was 31 July for each year. Maximum mean daily temperatures occur consistently during mid-July at EastGRIP. However, maxima in $\delta^{18}O$ have been observed to trail temperature maxima by as much as a
month at EastGRIP (Harris Stuart et al., 2023) likely due to post-depositional sublimation (Wahl et al., 2022). Similar patterns have been observed at Dome C, Antarctica, a much lower accumulation, but colder, site (Casado et al., 2018). We assign the peak summer $\delta^{18}O$ date assignment a $2\sigma$ uncertainty of $\pm 7$ days.

In assigning a date to the winter $\delta^{18}O$ minimum, it is important to recognize that the interior Greenland ice sheet can experience moderately coreless winters similar in character to the interior Antarctic ice sheet (Putnins, 1970; Schwerdtfeger, 1970).
So, the minimum mean monthly temperature may occur in any month from December through April. In addition, although diamond dust does occur on the Greenland ice sheet, most precipitation does not come during the minimum temperatures; the minimum $\delta^{18}O$ values represent the coldest *precipitation* events. We assume that these coldest precipitation events happen during the coldest months, but assigning a date to the coldest precipitation events overreaches the power of our meteorology data. So, we set the date for minimum $\delta^{18}O$ values to the first of each coldest month, resulting in a $2\sigma$ uncertainty of $\pm 15$ days.
The peaks in the snow profiles are not always in sharp relief from their neighbors. If we assume that the choice of the $\delta^{18}O$ maxima/minima values might be off by as much as one sample level in a snow profile, then the vertical sampling resolution results in an error of $\pm 1$ cm for the top 10 cm of each profile and $\pm 2$ cm for the rest of each profile. If accumulation rate is



approximately 40 cm·$y^{-1}$ of snow, then the resulting uncertainty in age-depth is approximately $\pm 9$ days for the top 10 cm and $\pm 18$ days for the rest each profile.

Altogether, we conservatively assess the $2\sigma$ uncertainty as a minimum of $\pm 9$ days for the top of each profile, $\pm 25$ days around each summer peak below 10 cm, $\pm 33$ days around each winter trough below 10 cm. During high accumulation rate time periods and events, the dating uncertainty will be much smaller, and vice versa. Surface height changes from PROMICE sonic ranger data (Fausto et al., 2021) indicates that accumulation rate at EastGRIP is not constant. Surface height changes are higher in summer and autumn than winter and spring, with approximately 50 percent of the surface height changes come from

20 percent of the observed events.

Figure 2 shows data from Transect 2 in 2019. Here, the depth adjustments provide a strong start for the age-depth model, and the age-depth model does not vary much from profile-to-profile. The age-depth model varies more between snow profiles taken during the 2017 season when the depth adjustment used to align profiles was not as strong.

The age-depth model is reliable when clear $\delta^{18}O$ maxima and minima exist in the snow profiles, which is true for the

majority of each profile. However, 10-20 percent of each profile remained unconstrained at the bottom of most cores because snow profiles rarely end in clear extrema. To date the lowermost ends of the snow profiles, we started with the earliest date assigned (i.e. deepest maxima or minima) and estimated the mean accumulation rate for the remaining snow from the sonic ranger data for that time period (Fausto et al., 2021). The age-depth model for this snow is the inverse of the mean accumulation rate. We manually quality-controlled the resulting $\delta^{18}O$ profile against the entire data set.

**A3   Missing data and other sources of uncertainty**

Transect line 4 was impacted by traffic or resampling during the 2017 field season. It was left out of these analysis.

Transect lines 2-5 were shifted inadvertently up one transect in the middle of the 2018 field season due to a change in field personnel. This was corrected during post-processing.

In addition to the 1-m profiles used here, nine shorter profiles (30 cm in length) were taken in 2017. As stated earlier, we do

not use these data here as they do not provide interannual information, are difficult to date, and would statistically overweight the top-of-core averages. Sampling of shallow profiles induced distance traveled along each transect in 2017, approximately 50 cm between each shallow profile. So, the total distance traveled along the 2017 transects is estimated as a conservative 13 m.

Compression often occurred during the extraction of the snow. Standard procedure would be to apply a correction for this compression evenly across each profile, particularly in deeper firn or ice. However, we believe that the location of compression

is more likely localized in near-surface snow (e.g. at fragile faceted layers). In a 1-m snow profile from this site, there are at least five locations where compression might have occurred, at the surface or the spring or autumn depth hoar layers. It is also certain that the compression did not occur evenly across any profile. The compression values are small relative to the profile lengths and identifying the hoar layers is difficult after extraction, and impossible after bagging and shipping. So, we leave the compression amount as an uncertainty in the dating, with a probable maximum value of 9 days.



Finally, we did not adequately assess the relative starting heights of the transects at the beginning of each season. This induces relative errors of around 3-5 cm in our depth adjustment between each snow profile based on May surface roughness estimates from Zuhr et al. (2021). The missing information does not impact the age-depth model.

**Appendix B: Tables of snow profile statistics**

Tables of statistics for the snow profiles and their changes presented in Figures 7c,d and 8c,d composited by season or year.

**Table A1.** Table of annual statistics for $\delta^{18}O$ from the EastGRIP snow profiles shown in Figure 7. Columns are the year of extraction, e.g. *2019* represents July-July annual average from snow extracted during the 2019 summer field season (also the dark blue curve in Figure 7c). Rows are the age of the snow. The annual cycle is winter-centric, and computed from 31 July to 31 July. Units are in $°/_{oo}$ and uncertainty is $2\sigma_{\bar{x}}$.

| *Extraction year* | *2017* | *2018* | *2019* |
|---|---|---|---|
| **Annual layer age** | | | |
| **07/2015 - 07/2016** | $-36.5 \pm 1.0$ | — | — |
| **07/2016 - 07/2017** | $-37.2 \pm 1.1$ | $-36.7 \pm 1.0$ | — |
| **07/2017 - 07/2018** | — | $-35.7 \pm 1.0$ | $-36.0 \pm 0.8$ |
| **07/2018 - 07/2019** | — | — | $-34.9 \pm 1.4$ |



**Table A2.** Table of changes in $\delta^{18}O$ concentration after one or two years of aging in the EastGRIP firn from Figures 7d and 7c, respectively. Columns are mean annual residuals, summer residuals (June/July), and non-summer residuals. Rows are the years between which the change is calculated. Units are in $^o/_{oo}$ and uncertainty is $2\sigma_{\bar{x}}$.

| | Annual | Summer | Non-summer |
|---|---|---|---|
| $\overline{\delta^{18}O}_{y2}$- $\overline{\delta^{18}O}_{y1}$ | | | |
| $\overline{\delta^{18}O}_{2018}$-$\overline{\delta^{18}O}_{2017}$ | $0.6 \pm 0.5$ | $0.3 \pm 0.3$ | $0.6 \pm 0.5$ |
| $\overline{\delta^{18}O}_{2019}$-$\overline{\delta^{18}O}_{2017}$ | $-0.9 \pm 0.6$ | $-1.6 \pm 0.3$ | $-0.7 \pm 0.5$ |
| $\overline{\delta^{18}O}_{2019}$-$\overline{\delta^{18}O}_{2018}$ | $-0.83 \pm 0.8$ | $-1.1 \pm 0.4$ | $-0.8 \pm 0.4$ |





**Table A3.** Table of annual statistics for $d$ from the EastGRIP snow profiles shown in Figure 7. Columns are the year of extraction, e.g. *2019* represents the black curve in Figure 8. Rows are the age of the snow. The annual cycle is winter-centric, and computed from 31 July to 31 July. Units are in $^o/_{oo}$ and uncertainty is $2\sigma_{\bar{x}}$.

| *Extraction year* | *2017* | *2018* | *2019* |
|---|---|---|---|
| **Annual layer age** | | | |
| **07/2015 - 07/2016** | $8.8 \pm 1.0$ | — | — |
| **07/2016 - 07/2017** | $8.9 \pm 1.1$ | $8.4 \pm 0.9$ | — |
| **07/2017 - 07/2018** | — | $8.5 \pm 1.0$ | $8.9 \pm 0.8$ |
| **07/2018 - 07/2019** | — | — | $9.0 \pm 1.4$ |





**Table A4.** Table of changes in $d$ concentration after one or two years of aging the EastGRIP firn from Figures 8a and 8c, respectively. Columns are mean annual residuals, summer residuals (June/July), and non-summer residuals. Rows are the years between which the change is calculated. Units are in $\permil$ and uncertainty is $2\sigma_{\bar{x}}$.

|  | *Annual* | *Summer* | *Non-summer* |
| --- | --- | --- | --- |
| $\bar{d}_{y2} - \bar{d}_{y1}$ |  |  |  |
| $\bar{d}_{2018} - \bar{d}_{2017}$ | -0.37 ± 0.4 | 1.11 ± 0.6 | -0.89 ± 0.4 |
| $\bar{d}_{2019} - \bar{d}_{2017}$ | -0.5 ± 0.4 | 4.3 ± 0.6 | -1.8 ± 0.4 |
| $\bar{d}_{2019} - \bar{d}_{2018}$ | 0.4 ± 0.4 | 3.3 ± 0.6 | -0.3 ± 0.4 |





**Table A5.** Table of statistics for $\delta^{18}O$ and $d$ from the EastGRIP surface snow, and $d$ from near-surface summer snow less than one year old, as shown in Figure 7 and 8. Columns are the isotopologues. Rows are the sampling time period. Units are in $^o/_{oo}$ and uncertainty is $2\sigma_{\bar{x}}$.

| | $\delta^{18}O_{sfc}$, summer | $d_{sfc}$, summer | $d$, summer snow profile, $<$ 1 year old |
|---|---|---|---|
| **Field season** | | | |
| **06-08/2016** | -27.7 $\pm$ 1.2 | 8.55 $\pm$ 1.5 | |
| **06-08/2017** | -31.28 $\pm$ 1.4 | 8.22 $\pm$ 2.9 | 7.76 $\pm$ 0.9 |
| **06-08/2018** | -32.19 $\pm$ 1.4 | 10.31 $\pm$ 2.5 | 5.41 $\pm$ 0.54 |
| **06-08/2019** | -26.39 $\pm$ 1.4 | 8.08 $\pm$ 2.4 | 3.72 $\pm$ 0.59 |



**Table A6.** Table of $\delta^{18}O$ vs $\delta D$ composited by age and season. summer is June-July. winter is December-April. Units are in $(^o/_{oo})/(^o/_{oo})$ and uncertainty is $2\sigma$.

|  | slope |
| --- | --- |
| **All data** | $8.05 \pm 0.003$ |
|  |  |
| **age $<$ 1 year** | $7.91 \pm 0.004$ |
| *Summer* | *7.87 $\pm$ 0.02* |
| *Winter* | *8.10 $\pm$ 0.01* |
| **1 year $\leq$ age $<$ 2 years** | $8.18 \pm 0.006$ |
| *Summer* | *8.56 $\pm$ 0.03* |
| *Winter* | *7.96 $\pm$ 0.02* |



**Table A7.** Table outlining assumptions and driving conditions behind the idealized modeling isotopic evolution of near-surface snow due to forced ventilation.

| | |
|---|---|
| Model snow | $\rho_{snow}$ = 350 kg m$^{-3}$; r$_g$ = 100 $\mu$m; $\kappa$ = 22 · 10$^{-10}$ m$^2$; sastrugi length = 1 m; sastrugi half-height = 0.1 m |
| Model atmosphere | $\rho_{atm}$ = 1.2 kg m$^{-3}$; viscosity of atmosphere = 1.2 kg·$m^{-1} \cdot s^{-1}$ |
| FV$_{dep}$ | $\delta^{18}O_{snw}$ = -30 ‰; $\delta^{18}O_{atm}$ = -40 ‰; T$_{snw}$ = -10 $^o$C; T$_{atm}$ = -15 $^o$C; $u$ = 5 m s$^{-1}$ -or- 10 m s$^{-1}$ |
| FV$_{dep}$ | $\delta^{18}O_{snw}$ = -30 ‰; $\delta^{18}O_{atm}$ = -40 ‰; T$_{snw}$ = -15 $^o$C; T$_{atm}$ = -10 $^o$C; $u$ = 5 m s$^{-1}$ -or- 10 m s$^{-1}$ |

## Appendix B:  Seasonal Isotope-to-temperature sensitivity

## Appendix C:  Supporting simulations

### C1   Isotope-gradient-driven diffusion simulation scenario

Johnsen et al. (2000) isotopic-gradient-driven (IGD) diffusion is used to explain the pattern and magnitude of the changes we observe in the near-surface snow at EastGRIP. The model is run on the mean $\delta^{18}O$ and $\delta D$ profiles from the 2019 field season using the following scenario that roughly approximates the annual cycle at EastGRIP: summer is 60 days with snow at -11$^o$C, autumn is 60 days with snow at -28.5$^o$C, winter is 180 days with snow at -40$^o$C, spring is 60 days with snow at -28.5$^o$C. This scenario is realistic, but may slightly overestimate the amount of diffusion due to the long warm summer used. Sensitivity tests find that applying the diffusion simulations to smoother mean profiles as opposed to individual profiles with sharper features underestimates the amount of isotopic-gradient diffusion.

### C2   Forced ventilation of near-surface snow simulation scenarios

Forced ventilation is simulated based on the snow and atmospheric conditions for the summer season at EastGRIP using an augmented version of Town et al. (2008b). We have added fractionation on sublimation and $\delta D$ to the model. The snow surface structure in the model is parameterized as rolling undulations as prescribed by Colbeck (1997), using peak-to-peak lengths of 1 m, and half-heights of 0.1 m, which reflect summertime observations from EastGRIP (Zuhr et al., 2021, 2023). Snow density is taken from (Komuro et al., 2021). The other snow parameters are reasonable assumptions for polar snow (Town et al., 2008b). The summertime isotopic content of atmospheric water vapor over the Greenland Ice Sheet was estimated based on measurements from NEEM (Steen-Larsen et al., 2014) and EastGRIP (personal communication, K. Rozimarek, 2023).

In addition to the simulation shown in Figure 9, two more simulations of the isotopic impact of forced ventilation on near-surface snow were performed. The first scenario uses wind speeds of 10 m/s, a typical cyclonic event over EastGRIP that may



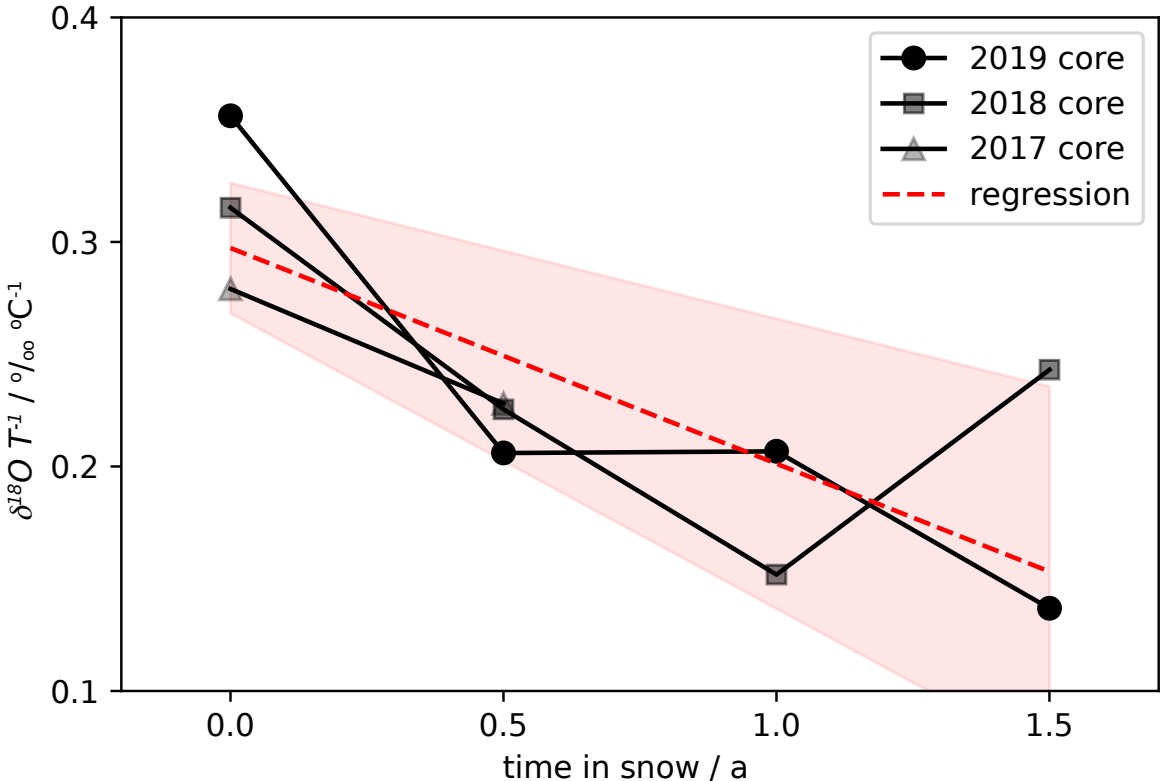

**Figure B1.** The increase in seasonal isotope-to-temperature sensitivity ($\gamma_t$), a decreasing slope, from EastGRIP based on one-and-a-half years of seasonal maxima and minima in $\delta^{18}O$ and mean monthly temperature. The regression is $2\sigma$ and accounts for errors in both variables. Here the line of best fit is $\Delta\delta^{18}O \, \Delta T^{-1}$= -0.096 $\pm$ 0.036 $^o/_{oo}$ $^oC^{-1}$ $a^{-1}\cdot$ (time in snow) + 0.297 $\pm$ 0.029 $^o/_{oo}$ $^oC^{-1}$.

last 3 days. The second scenario uses wind speeds of 20 m/s, a strong cyclonic event at EastGRIP that may last 24 hours. During these sorts of events the snow surface is dynamic. We do not take this into account in these simulations.



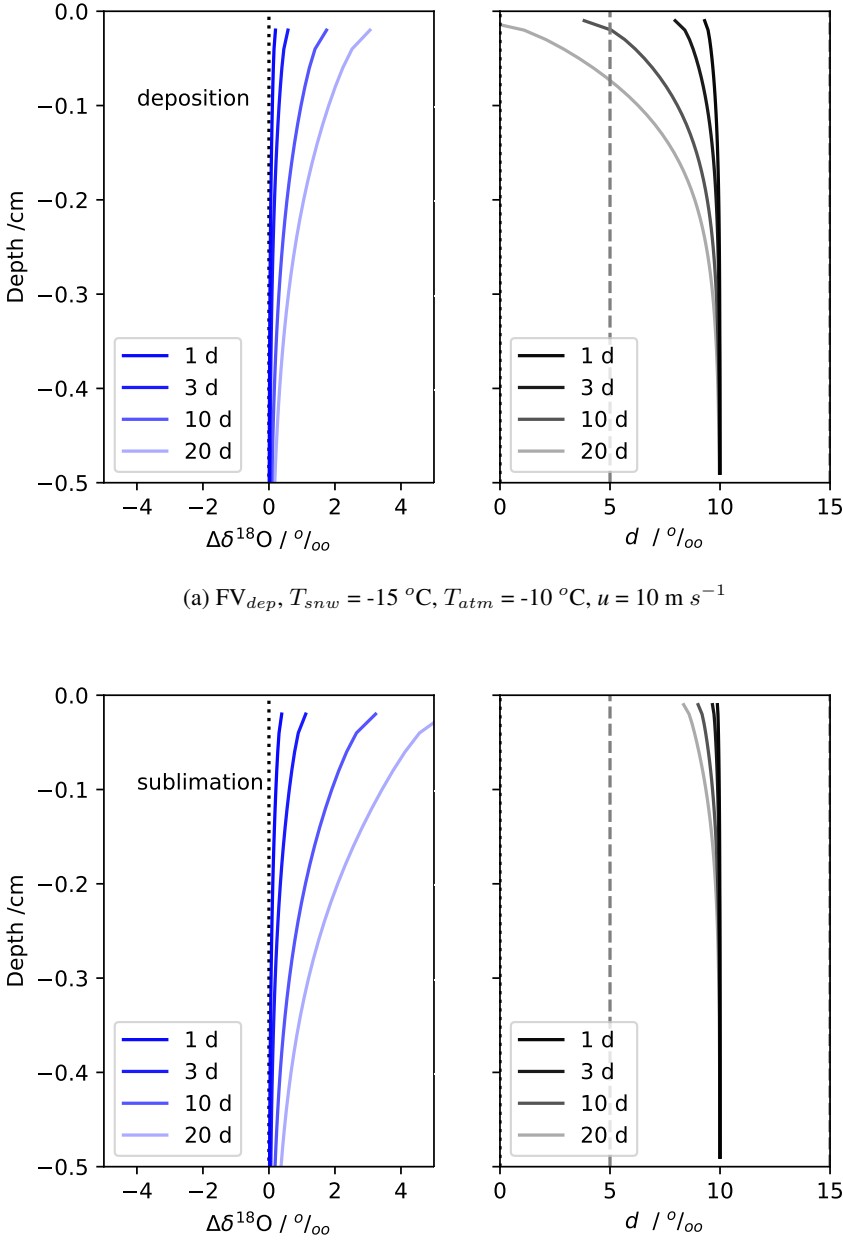

(a) FV$_{dep}$, $T_{snw}$ = -15 $^o$C, $T_{atm}$ = -10 $^o$C, $u$ = 10 m $s^{-1}$

(b) FV$_{sub}$, $T_{snw}$ = -10 $^o$C, $T_{atm}$ = -15 $^o$C, $u$ = 10 m $s^{-1}$

**Figure A1.** Idealized simulations of forced ventilation on isotropic snow under high wind EastGRIP summertime conditions. The snow structure and isotopic content of snow and vapor are detailed in Table A7.



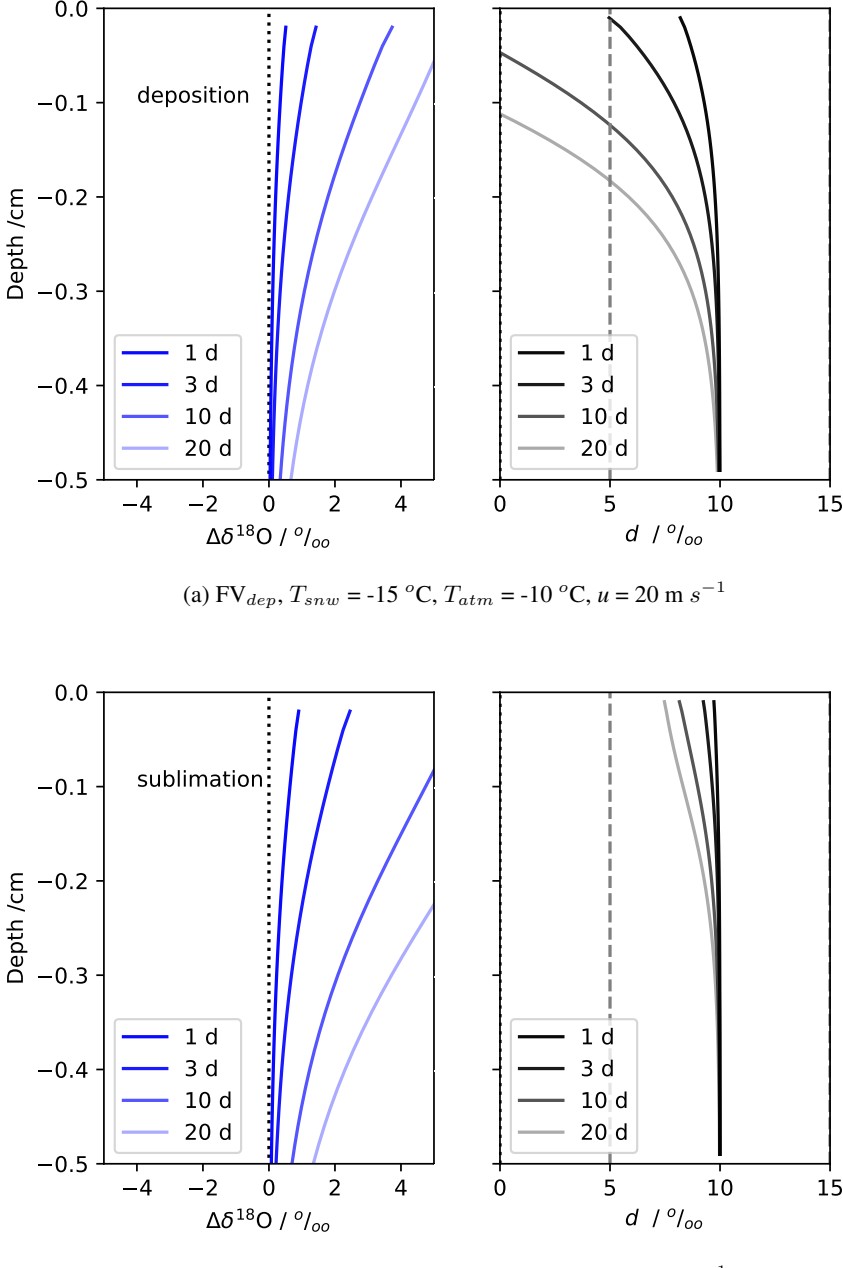

(a) $FV_{dep}$, $T_{snw}$ = -15 $^o$C, $T_{atm}$ = -10 $^o$C, $u$ = 20 m $s^{-1}$

(b) $FV_{sub}$, $T_{snw}$ = -10 $^o$C, $T_{atm}$ = -15 $^o$C, $u$ = 20 m $s^{-1}$

**Figure A2.** Idealized simulations of forced ventilation on isotropic snow under extremely high wind EastGRIP summertime conditions. The snow structure and isotopic content of snow and vapor are detailed in Table A7.



*Author contributions.* MST and HCSL designed the study. HCSL, AKF, SW, and MB obtained the observational dataset. MB and AS analyzed the snow samples. MST and MB curated and processed the snow profile data set. MST performed the formal analysis and wrote the manuscript. HCSL, TJ, and SW contributed to the interpretation of the analyses. Reviews and edits were made by all co-authors. HCSL designed and acquired funding for this study and administrated the SNOWISO project.

*Competing interests.* There are no competing interests present in this work.

*Acknowledgements.* This project received the majority of funding from the European Research Council (ERC) under the European Union's Horizon 2020 research and innovation program: Starting Grant- SNOWISO (grant agreement 759526). The simulations and some manuscript writing were funded by Collaborative Research: NSFGEO-NERC: Integrated Characterization of Energy, Clouds, Atmospheric state, and Precipitation at Summit: Measurements along Lagrangian Transects (Grant number 2137083).

EastGRIP is directed and organized by the Centre for Ice and Climate at the Niels Bohr Institute, University of Copenhagen. It is supported by funding agencies and institutions in Denmark (A. P. Møller Foundation, University of Copenhagen), USA (US National Science Foundation, Office of Polar Programs), Germany (Alfred Wegener Institute, Helmholtz Centre for Polar and Marine Research), Japan (National Institute of Polar Research and Arctic Challenge for Sustainability), Norway (University of Bergen and Bergen Research Foundation), Switzerland (Swiss National Science Foundation), France (French Polar Institute Paul-Emile Victor, Institute for Geosciences and Environmental research), and China (Chinese Academy of Sciences and Beijing Normal University).

Many talented workers helped in the collection and measurement of the snow. Basile de Fleurian, Johannes Freitag, Abby Hughes, Emma Kahle, Martin Madsen, Hannah Meyer, Silje Smith-Johnsen, Alexandra Touzeau, Diana Vladimirova, and Tobias Zolles assisted in the field with snow collection and processing. Þorsteinn Jónsson and Rósa Ólafsdóttir assisted in the isotopic measurement of snow. Laura Dietrich shared with us her version of the Johnsen et al. (2000) isotopic-gradient-diffusion model.

Von Walden, Maria Hörhold, and Alexandra Zühr helped revise the manuscript.



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
