# Peer review of "Post-depositional modification on seasonal-to-interannual timescales alters the deuterium excess signals in summer snow layers in Greenland"

_EGUsphere, 2023_

## Author Response (AR1)

**Response to reviews for EGUSPHERE-2023-2462**

In ourresponse to reviews, we have indicated that we revised the manuscript according to the reviewers' comments. We have also edited some small things for clarity along the way. Please contact me with any questions regarding our response to the reviewers' comments.

Please note, my mark-up tool (diffpdf) compares page-by-page. So, where a section was edited (Section 4.1), the pages are shifted and they look completely changed, but are rather only slightly changed.

Please contact us with any questions, comments, or concerns.

Sincerely,

Mike Town (on behalf of all authors)

**From the authors (RC1):**

Referee 1 provides a clear and concise review. We agree with have taken action on all comments below. In particular, Referee 1 asks a question below about why there is a change in d-excess under equilibrium fractionation. This question spurred inclusion of kinetic fractionation in our modeling of forced ventilation. The new model runs have been incorporated into the manuscript, with the equilibrium model runs retained in the appendix for readers' reference. This represents the only important change to the manuscript based on Referee 1's comments. More specific responses to Referee 1 comments are below.

Thank you for this review.
* * *
Referee 1:

The manuscript by Dr. Town and others continues the series of publications of the EGRIP team devoted to the study of the post-depositional changes in the isotopic composition of the surface and near-surface snow. In this manuscript the results of the field experiments are shown that are aimed at observing "in real time" the evolution of the oxygen 18 and dxs values in given snow layers during a summer season and between summers.

Such studies are strongly needed to understand how the climatic signal is modified between a precipitation event and a "lock in" of this signal in an ice core.

I have only a few minor comments to the manuscript.

| Reviewer comment | Response |
|---|---|
| Line 6: "The d always experiences a 3-5 ‰ increase in d" – the last "in d" is redundant. | Yes, thank you. |
| Lines 12-13: "Finally, we observe a dramatic increase in the seasonal isotope-to-temperature sensitivity occurs, which can be explained solely by IGD diffusion" – "occurs" is unnecessary. | Yes, thank you. |
| Table 1 – does the limit of 1 cm for the surface snow's lower boundary is defined somewhere? Or is it just related to the sampling procedure? In the previous papers (e.g., Wahl et al., 2022) I remember the surface sampling meant sampling of the upper 0.5 cm. | This limit is due to the sampling procedure. Snow samples were made at a range of integrated depths. But, your comment has highlighted that we need to adjust the references for this data set. Thank you. |
| Line 47: "The critical distinctions here are if the it precipitation." – sorry, did not understand this. | No apology necessary. In fact, we should apologize as this 'sentence' was unintelligible. We have removed it as the previous paragraph and the Table suffice. |
| Line 48: "we refer to is as" – to it? | This paragraph has been removed. |

| | |
|---|---|
| Line 147 – is it important that it is a national park in the context of the paper? More important that it's an ice sheet. But ok, If you like so. | We have removed reference to the National Park |
| Lines 200-204 – what is the uncertainty of dxs? | Good question. Propagating errors in eq 2, we see that dxs error is ~2 per mille. We have added this to the text. |
| Lines 273-274: "The potential influence of forced ventilation on near-surface snow due to tapers off dramatically after about 50 cm" – it looks like something is missed in this sentence. | Thank you. We have revised this paragraph.
"The potential influence of forced ventilation tapers off dramatically after about 50 cm depth in near-surface snow." |
| Figure 7 – panel b is empty. | We have filled it. Thank you. |
| Line 367 – useful to remember? | We have revised this to start with "The phenomenon of post-depositional..." |
| Line 429-430 – why do you assume equilibrium? In the Wahl et al., 2022 paper they show that kinetic fractionation much better explained the isotopic modification of the surface snow. Ok, I see the answer lower in the text (471-472). | Based on the question below, we have added kinetic fractionation to the model and reran all scenarios with this fractionation scheme. We have included one equilibrium fractionation scheme in the Appendix for the readers' reference. |
| Line 442 – if you assume equilibrium fractionation, then why does dxs change? | This was an excellent question. It provoked an augmentation of the model. The reason there was a d-excess change was due to the linear definition of d-excess although the relationship is more properly modeled as non-linear at cold temperatures. ( Duetsch 2017 (10.1002/2017JD027085) ) |

| | To remedy this under the constraints of the review process, we added kinetic fractionation to the ventilation model. This produced results under deposition consistent with our observations. The sublimation results changed in magnitude, but not in character. Figures 9, C1, and C2 have changed accordingly. We have moved the equilibrium fractionation simulation (formerly Figure 9) to Figure C3 for the readers' reference.

Section 4 has changed slightly in reaction to the results for deposition under kinetic fractionation.

Thank you for this question! |
|---|---|
| Section 4.1.5 – is it really needed here? Your paper is devoted to another topic, not to the accumulation rate. | You are correct, this section does seem out of place. We have revised the inclusion accumulation information to be less jarring. |
| Lines 533-534: "7.87 o/oo · o/oo−1" – what is this dot between ‰ and ‰? | This dot represents a multiplication. We have chosen a different representation of the ratio of per mille (permille)-1. |
| Line 543: "It seems logical that the large reservoir of the atmosphere because" – something seems to be missed in this phrase. | The mechanisms at work interannually are then likely a combination of IGD diffusion and other post-depositional processes.
We have revised this sentence to say:

'It seems logical that because the atmosphere is such a large potential reservoir of vapor, we are looking for processes that bias the isotopic signal towards an increase in $d$. |
| Lines 544-545: "causes a negative Δd during summer through surface sublimation but possibly | Corrected. Thank you.

" Our simulations indicate that the near-surface atmosphere likely causes a negative \Ddxs\ during summer |

| | |
|---|---|
| through force ventilation of near-surface snow" – the same. | through surface sublimation but possibly also through force ventilation of near-surface snow." |
| Line 557: "We suggest is to dividing the contribution" – better "we suggest to divide the contribution"? | Corrected, thank you.

'We suggest dividing the contribution' |

**From the authors (RC2):**

The referee's comments are relevant and helpful in every respect. Our thanks to the referee for this cogent review.
* * *
Referee 2 (Casado):

**Review of « Post-depositional modification on seasonal-to-interannual timescales alters the deuterium excess signals in summer snow layers in Greenland» by Town and others.**

This manuscript describes measurements of short snow pits gathered at EASTGRIP, in Greenland. The authors use the variation in snow isotopic composition after the deposition took place to study the post deposition processes that alter the isotopic composition of near surface snow. Specifically, they consider the impact on the d-excess of these post deposition processes in summer due to latent heat flux near the surface.

This work relies on a new set of short pits that they were obtained across a few field seasons as well as an exhaustive dataset of meteorological measurements. A depth adjustment has been implemented to align the profiles taking into account the elevation offset due to the accumulation during the years, as well as the spatial heterogeneity of snow accumulation in Polar Regions. This provides the authors with comparable isotopic profiles where they can evaluate the change of isotopic composition after the snow has been deposited. The authors quantified whether the changes were statistically significant, but some improvements on this aspect would be beneficial to the manuscript. Modelling of the effects of post-deposition processes is realised to support the qualitatively the observations both at the seasonal and interannual scale.

The manuscript is well written, albeit possibly too long. Some elements in the figure were missing. Apart from a couple of minor comments, I would recommend to accept the manuscript for publication.

| Reviewer comments | Responses | |
|---|---|---|

| | | |
|---|---|---|
| Line 7: I don't think citations are supposed to be present in abstracts. | You are correct. Thank you. | x |
| Line 195: "The snow was cut in an open-faced tray using a 0.10-cm thick blade."

It should be "0.1cm", or "1mm". | The precision of this cutting tool is sub-millimeter. But the cutting process is a bit uncertain in a relative sense, so we will use 1-mm. | x |
| Line 309: "Significant increases ($p < 0.05$) in $\delta 18O$ are seen the summers of 2017 and 2019, down to 20-30 cm."

It seems to me in these figures that the increases are really near the surface, like the first 10-15cm. Also, this might be explained later, but how are these increase significant while most of the bottom part of the cores are decreasing? This should be mentioned here, as the increase in the first 10 cm is more or less of the same amplitude than the decrease between 30 and 50 cm in Figure 4. | We will revise this claim so as not to overstate what is present in the data. This will not change the analysis, but the related analysis has also been revised slightly. | x |
| Figure 7: Panel b is empty. | This has been fixed. | x |
| Line 418: "We see an attenuation in peak $\delta 18O$ of up to 2 o/oo due to IGD diffusion (See Figure 11)." | Apologies. This has been changed to 'not shown'. | x |

| | | |
|---|---|---|
| There is no d18O in Figure 11. | | |
| Line 558: "The historical approach has been to consider the atmosphere as the source of the climate signal, and the deep firn as a source of noise to be inverted"

I don't think noise is the right term, as diffusion, advection and thinning are not adding noise to the signal, but acting respectively as low pass filter, phase modulation, and z-axis compression. I guess more as a "transfer function". | We have revised this language to read: The historical approach has been to consider the atmosphere as the source of the climate signal, and the deep firn as sources of processes to be inverted to resolve the climate signal. | x |
| Line 560: "As stated earlier, the atmosphere is often represented by climate-to-isotope sensitivities that reduce to temperature-to-isotope sensitivities,"

      This is nonsensical. | It is common 'vernacular' in paleoclimate and other related communities to use the words 'climate' and 'temperature' of a location/region synonymously. In this sentence, we are intending a direct contrast between these terms. 'Climate' includes many more things besides temperature (e.g. winds, RH, SEB). In this claim, we are stating that there are many more climatic influences on isotopic content of snow than simply local or regional temperature, but they are ignored. Rather, the various climate influences on isotopic content of snow other than temperature are misattributed (or reduced) to temperature.  We have clarified this line in the text as follows:

As stated earlier, the \emph{atmosphere} is often represented by climate-to-isotope sensitivities that reduce to temperature-to-isotope sensitivities, assuming all climate variability is represented by temperature or precipitation-weighted temperature. This often reducing further | x |

| | | |
|---|---|---|
| | to empirical linear relationships between spatial or temporal temperature data sets and \dO. A range of physically-nuanced models are used for dealing with the \emph{atmosphere} category more explicitly \citep{Werner2011, Dee2015, Hu2022, Markle2022}, but the overall approach to climate reconstruction is the same. | |
| Conclusion: Considering the extent of the discussion, I would recommend shortening the conclusion. | Good idea. We have done so. | x |

**Minor Comments:**